# Senescent cells perturb intestinal stem cell differentiation through Ptk7 induced noncanonical Wnt and YAP signaling

Jina Yun[1], Simon Hansen[2], Otto Morris[3], David T. Madden[4], Clare Peters Libeu[4], Arjun J. Kumar [5], Cameron Wehrfritz[4], Aaron H. Nile [6], Yingnan Zhang[1], Lijuan Zhou[1], Yuxin Liang[1], Zora Modrusan[1], Michelle B. Chen[1], Christopher C. Overall[1], David Garfield[1], Judith Campisi [4], Birgit Schilling [4], Rami N. Hannoush [1] ✉ & Heinrich Jasper [1,4] ✉

Cellular senescence and the senescence-associated secretory phenotype (SASP) are implicated in aging and age-related disease, and SASP-related inflammation is thought to contribute to tissue dysfunction in aging and diseased animals. However, whether and how SASP factors influence the regenerative capacity of tissues remains unclear. Here, using intestinal organoids as a model of tissue regeneration, we show that SASP factors released by senescent fibroblasts deregulate stem cell activity and differentiation and ultimately impair crypt formation. We identify the secreted N-terminal domain of Ptk7 as a key component of the SASP that activates non-canonical Wnt / $Ca^{2+}$ signaling through FZD7 in intestinal stem cells (ISCs). Changes in cytosolic $[Ca^{2+}]$ elicited by Ptk7 promote nuclear translocation of YAP and induce expression of YAP/TEAD target genes, impairing symmetry breaking and stem cell differentiation. Our study discovers secreted Ptk7 as a factor released by senescent cells and provides insight into the mechanism by which cellular senescence contributes to tissue dysfunction in aging and disease.

Cellular senescence is recognized as an evolutionarily conserved process in tumor suppression, wound healing and regeneration. However, the accumulation of senescent cells during aging can contribute to age-related diseases[1–8]. Cellular senescence is characterized by an essentially permanent cell cycle arrest, as well as by impaired apoptosis[9–12]. A critical feature of senescent cells is the senescence-associated secretory phenotype (SASP): the secretion of numerous proinflammatory cytokines, chemokines, proteases, growth factors, and lipids[13–17]. SASP factors have been implicated in promoting cellular plasticity and stem cell gene expression[18], but can also cause inflammation and negatively impact the local tissue environment and function. Whether and how the SASP influences stem cell function and thus impacts tissue regeneration in the aging organism, however, remains unclear.

Organoid systems provide an efficient and convenient model of tissue regeneration to dissect the effects of SASP components on stem cell function. Intestinal organoids, for example, can be established from single stem cells that develop into budding organoids with defined crypt and villus compartments[19,20]. During organoid development, stem cells proliferate and form a multicellular sphere, which breaks symmetry and self-organizes into a crypt-containing organoid[21]. Fully mature intestinal organoids recapitulate cell types and features present in the intestinal epithelium, providing an attractive system for disease modeling and drug screening[22–30].

[1]Genentech, Inc., 1 DNA Way, South San Francisco, CA 94080, USA. [2]NBE Therapeutics, Hochbergstrasse 60C, 4057 Basel, Switzerland. [3]Exscientia Ltd., The Schrödinger Building Oxford Science Park, Oxford OX4 4GE, UK. [4]Buck Institute for Research on Aging, 8001 Redwood Blvd, Novato, CA 94945, USA. [5]Fred Hutch/University of Washington, 1100 Fairview Ave. N., Seattle, WA 98109, USA. [6]Calico Labs LLC., 1170 Veterans Blvd, South San Francisco, CA 94080, USA. ✉e-mail: ramih@gene.com; jasperh@gene.com

The Wnt signaling pathway is critical for intestinal stem cell proliferation and maintenance[31–35]. Mutations causing constitutive activation of Wnt signaling are associated with epithelial hyperplasia and tumor formation, whereas reduced Wnt signaling, for example due to elevated expression of the negative Wnt regulator Notum in intestines of old mice, leads to decreased stem cell activity and impaired regeneration[36–38].

Canonical β-Catenin-dependent Wnt signaling is triggered by Wnt ligand-mediated formation of a ternary complex between Fzd receptors and LRP5 or 6 co-receptors, resulting in inhibition of the β-catenin destruction complex[32,39]. β-catenin-independent Wnt signaling, sometimes referred to as noncanonical Wnt signaling, can be triggered by Wnt ligand binding to Fzd receptors with co-receptors Ror2, Ryk, and Protein Tyrosine Kinase 7 (PTK7), leading to activation of JNK and Ca$^{2+}$ signaling[40–43]. Noncanonical Wnt signaling results in cytoskeletal rearrangements and influences cell migration and planar cell polarity. While the role of canonical β-Catenin-dependent Wnt signaling in intestinal stem cell activity and regeneration is well established, it remains unclear whether β-catenin-independent Wnt signaling influences intestinal stem cells.

Both noncanonical and canonical Wnt signaling can regulate the transcriptional coactivators YAP/TAZ[44–47]. Upon activation, YAP/TAZ proteins relocate to the nucleus where they form a complex with TEAD transcription factors to regulate cell proliferation, cell polarity, and other cellular functions[48–50]. In the intestine, YAP-mediated expression of a fetal gene signature has been implicated in epithelial regeneration[51–54]. In intestinal organoids, heterogenous YAP activation in stem cells is critical for symmetry breaking during early organoid development[21].

Ptk7, also called colon cancer Kinase 4 (CCK4), is a member of the receptor tyrosine kinase (RTK) family, and contains N-terminal Immunoglobulin-like repeats (Ig), a transmembrane region, and an inactive tyrosine kinase domain[55–57]. The extracellular domain of Ptk7 can be shed after cleavage by metalloproteases, and the kinase domain then translocates to the nucleus after subsequent cleavage by γ-secretase[58,59]. Ptk7 influences cellular processes such as migration and planar cell polarity (PCP)[60–67]. In *Drosophila*, the Ptk7 orthologue Otk regulates intestinal stem cell migration after epithelial injury. Injury-induced expression of matrix metalloproteinases promotes shedding of the N-terminal domain of Otk from enteroendocrine cells, inducing noncanonical Wnt signaling in ISCs, and promoting their migration towards the wound[68].

Here, we report the identification of the N-terminal domain of Ptk7 as a SASP factor elevated in aging intestines, and released from senescent cells. In mouse intestinal organoids, N-terminal Ptk7 causes defective differentiation of intestinal epithelial cells. We find that secreted Ptk7 acts through Fzd receptors on intestinal stem cells (ISCs) to activate Wnt/Ca$^{2+}$ signaling, which in turn triggers nuclear translocation of YAP. Changes in YAP/TEAD-dependent gene expression are responsible for Ptk7-induced defects in ISC differentiation. Our work identifies Ptk7 secreted by senescent cells as a critical mediator of ISC dysfunction.

## Results

### SASP impairs cell differentiation in mouse intestinal organoids

To ask whether and how SASP factors can influence ISC function and regeneration, we exposed mouse intestinal organoids derived from isolated crypts of the small intestine (jejunum) to conditioned media (CM) from mouse embryonic fibroblasts (MEFs) that were induced to senesce by X-ray irradiation or doxorubicin treatment (Fig. 1a[69,70],). Senescence-associated beta-galactosidase (SA-β-gal) staining confirmed senescence in the MEFs (Supplementary Fig. 1a). As a control, we used CM from quiescent (serum deprived) MEFs, which were growth arrested, similar to senescent cells. Organoids from fresh crypts acquired a distinct spherical cyst morphology after exposure to senescent CM (SCM), while organoids exposed to quiescent CM (QCM)

exhibited normal crypt budding (Fig. 1b, c). Cystic organoids lacked defined crypt domains (Fig. 1d). Immunostaining for Ki67 shows significant proliferative activity in organoids exposed to both QCM and SCM. Mitotic figures (pH3+ cells) were observed mainly in crypt domains of QCM organoids, but were distributed throughout all areas of SCM organoids (Fig. 1d). These observations suggest that exposure to SCM did not cause secondary senescence in organoids (as no sign of cell cycle arrest was observed based on pH3 and Ki67 staining), but influence ISC function in different ways (Supplementary Fig. 1b, c). To assess if the cystic organoid phenotype entails changes in cell composition, we compared transcript levels of various cell-type specific markers in QCM and SCM organoids. We observed a strong reduction in *OLFM4* (intestinal stem cell), *CHGA* (enteroendocrine cell) and *MUC2* (goblet cell) transcript levels in SCM organoids, while *ALPI* (enterocyte), *MKI67* (proliferating cell), *LGR5* (stem cell) and *LYZ* (Paneth cell) transcripts were not significantly changed (Fig. 1e). Whole mount immunostaining confirmed these findings: secretory cells such as goblet, enteroendocrine and Tuft cells (Dclk), and Olfm4+ SCs were reduced, while other cell types, including enterocytes, were not (Fig. 1f–m, Supplementary Fig. 1d).

We confirmed these findings using single-nucleus RNA sequencing of organoids exposed to QCM and SCM. Using a droplet-based single-nucleus method, we sequenced 38154 nuclear transcriptomes from QCM and SCM-exposed organoids. Unsupervised graph-based clustering identified 15 distinct clusters, which were annotated using known cell markers (Fig. 1n, Supplementary Fig. 1e). The relative contribution of cells from QCM and SCM exposed organoids to each cluster revealed differences in cell-type composition between QCM and SCM exposed organoids (Fig. 1o, p, and Supplementary Fig. 1f): Compared to QCM exposed organoids, SCM exposed organoids consist of an increased proportion of stem cells and of cells expressing genes of a fetal signature, but a decreased proportion of enteroendocrine progenitors and of differentiated cells of the secretory lineage such as enteroendocrine, Tuft, and Paneth cells.

Critically, the morphological change caused by SASP factors was reversible (Fig. 1q, Supplementary Fig. 1g). When organoids were switched from QCM to SCM 4 days after seeding (when crypt buds become visible in QCM), they were smaller and exhibited less budding and cystic crypt domains. Budding organoids did not become completely cystic, suggesting that SASP factors influence stem cells primarily at early stages of symmetry breaking. On the other hand, cystic organoids switched from SCM to QCM 4 days after seeding, started budding and lost cystic morphology. Consistently, we observed no long-term effect of SCM on stem cell activity. When organoids exposed to SCM were passaged and maintained in regular organoid culture media, colony-forming efficiency remained comparable to those exposed to QCM (Supplementary Fig. 1h, i). Together, these data indicated that SCM causes specific and reversible defects in cell differentiation in the intestinal stem cell lineage.

### Secreted Ptk7 is a SASP factor causing the cystic organoid phenotype

SASP factors include proinflammatory cytokines, chemokines and growth factors[9–12]. To identify factor(s) in SCM responsible for the morphological changes in organoids, we analyzed proteins in SCM by mass spectrometry (MS). SCM from a large-scale culture of senescent MEFs was dialyzed, concentrated and fractionated by ion exchange and size exclusion columns (Fig. 2a). Fractions categorized as containing small, medium and large proteins were tested for cystic organoid inducing activity, which were present only in the medium fraction (Supplementary Fig. 2a). MS analysis of the small and medium fractions identified and quantified a total of 148 proteins (39 differentially present proteins with the following cutoffs: $q$ value <0.05, absolute fold change >1.5) (Fig. 2b, Supplementary Fig. 2b, Supplementary Data 1 and 2). Proteins previously described as SASP components were

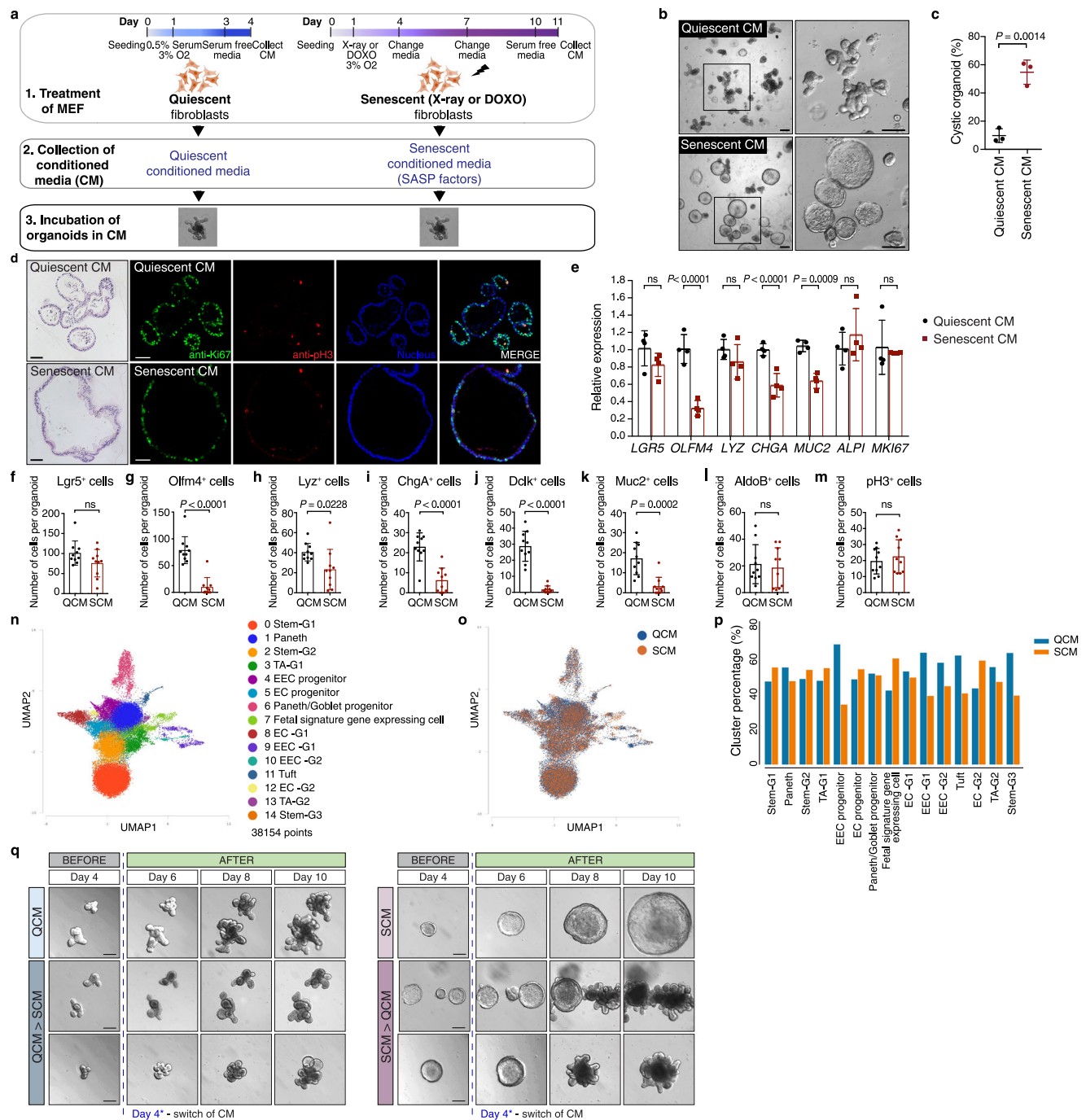

**Fig. 1 | Conditioned medium from senescent fibroblasts causes cystic morphology of mouse intestinal organoids. a** Schematic showing induction of quiescence and senescence in fibroblasts, conditioned media collection, and organoid culture in conditioned media. The figure was created with BioRender.com. **b** Representative images of organoids after 5 d of culture in conditioned media. Enlarged images (boxed area) are shown at the right. Scale bar, 200 μm. **c** Quantification of data from (**b**). The percentage of cystic organoids in organoids cultured in conditioned media is shown (mean ± s.d.; two-tailed t-test). Each dot represents a result from organoids established from a mouse, 3 biological replicates per experiment (n = 3). **d** H&E staining (left column) and immunostaining of paraffin-embedded organoids. Immunostaining for Ki67 (green) and phospho-Histone H3 (red) in intestinal organoids cultured in conditioned media and stained with Hoechst (blue). Scale bar, 50 μm. **e** Gene expression in organoids exposed to SCM determined by qPCR and represented as a ratio relative to organoids exposed to QCM (mean ± s.d.; n = 4; two-tailed t-test). **f**–**m** Numbers of cells expressing the listed markers in organoids exposed to QCM or SCM (mean ± SEM; n = 10; two-tailed t-test). **n** UMAP of pooled snRNA-seq data of nuclei from QCM and SCM exposed organoids colored by cell-type annotation (right). **o** UMAP showing contribution of nuclei from QCM and SCM exposed organoids labeled in blue and orange, respectively. **p** Relative contribution of cells from QCM and SCM exposed organoids to each cluster. Cells from QCM and SCM exposed organoids were labeled in blue and orange, respectively. **q** Representative images of organoids taken every 2 days starting on day 4. From day 0 to 4, organoids were cultured in quiescent conditioned media (left panel) and senescent conditioned media (right panel). On day 4, quiescent-conditioned media was changed to senescent-conditioned media (left bottom panel), whereas senescent-conditioned media was changed to quiescent-conditioned media (right bottom panel). In all conditions, media was changed every other day. Scale bars, 100 μm. See also Fig. S1. Source data are provided as a Source Data file.

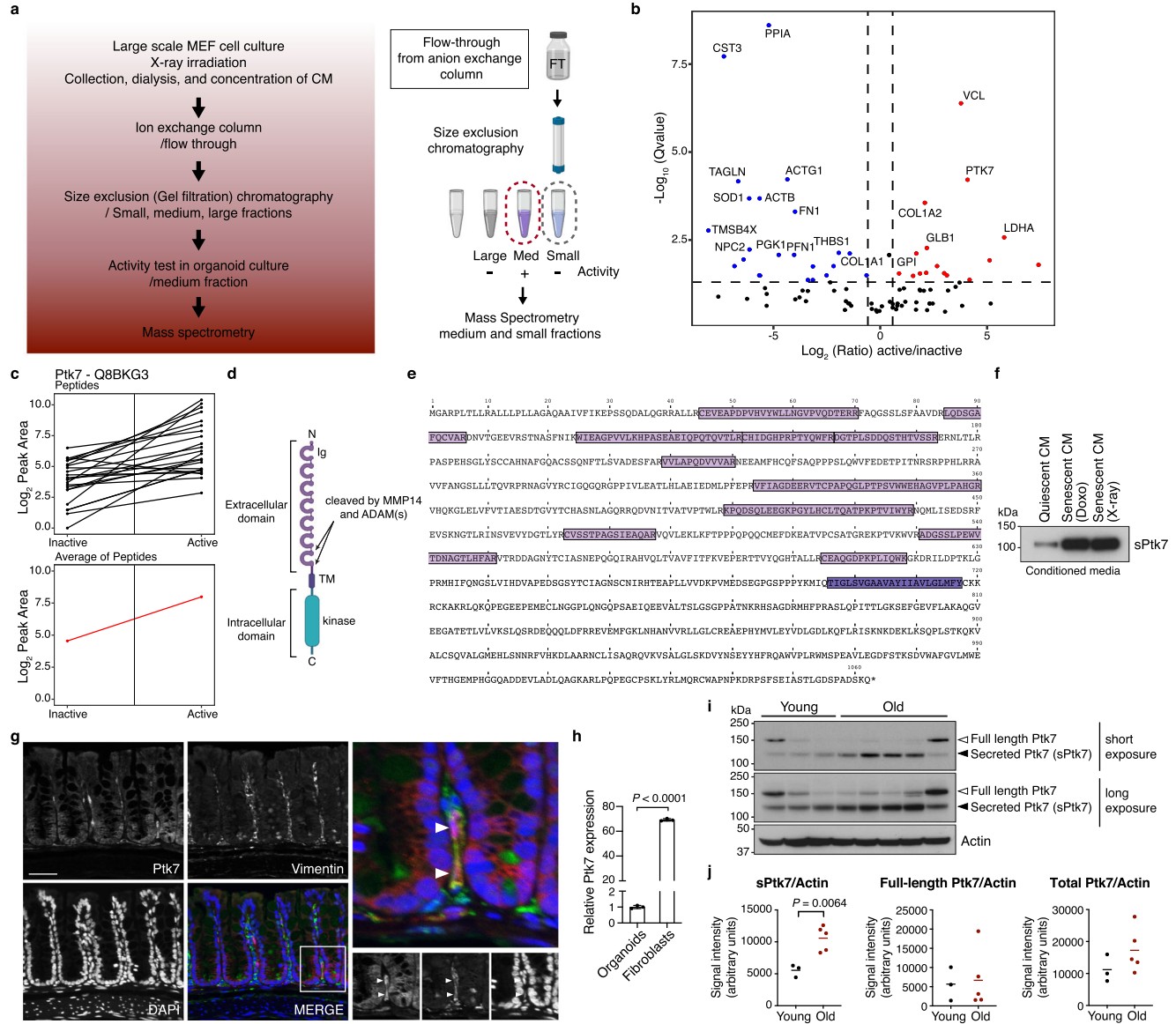

**Fig. 2 | Identification of Ptk7 as a factor that causes cystic organoid phenotype.** **a** Schematic showing conditioned media preparation, column purification, activity test, and mass spectrometry (MS). The figure was created with BioRender.com. **b** Volcano plot representing the statistical significance and protein abundance ratios between medium (active) and small (inactive) fractions. Dashed lines depict a $q = 0.05$ cutoff (horizontal) and 1.5-fold difference cutoffs (vertical). Proteins that are present highly in medium (active) and small (inactive) fractions are represented in red and blue, respectively. Proteins with $q < 0.01$ and more than 1.5-fold difference are labeled. **c** Graphs showing peptide peak area profiles for Ptk7. Each Ptk7 peptide detected is shown as a black line, indicating the peak area measurements for the inactive and active fractions. The average of all Ptk7 peptides is presented as a red line. **d** Schematic representation of the Ptk7 protein domain structure. Ig: Immunoglobulin-like loops (lilac), TM: a transmembrane region (purple), and an inactive tyrosine kinase domain (blue). **e** A graphic representation of the list of peptides detected by MS. Ptk7 peptide sequences detected by MS in lilac and the

transmembrane domain in purple. **f** Western blot for Ptk7 in conditioned media collected from quiescent and doxorubicin- or irradiation-induced senescent MEFs. Conditioned media corresponding to the same number of cells was loaded. Increased shedding of Ptk7 (soluble form of Ptk7; sPtk7) was detected in senescent conditioned media. **g** Immunofluorescence staining of Ptk7 (red) and stromal cell marker Vimentin (green) in the large intestine. Nuclei were counter stained with DAPI (blue). Scale bar: 50 μm. **h** The relative *PTK7* expression in intestinal organoids and fibroblasts (mean ± s.d.; $n = 3$; two-tailed $t$-test). **i** Western blots of Ptk7 in the large intestine extract from young and old mice. Blot images of two different exposure times are shown. Each lane presents tissue lysate from one mouse ($n = 3$ for young mice, $n = 5$ for old mice). Actin was used as loading control. **j** Quantification of data from (**i**). Ptk7 and Actin protein levels were analyzed by ImageJ. Ptk7 protein levels normalized to Actin are shown as mean ± s.d. Two-tailed $t$-test was used for statistical analysis. See also Fig. S2. Source data are provided as a Source Data file.

identified in these SCM fractions, including CCL2 and IL1RL1 (Supplementary Data 2 and 3, SASP Atlas, http://www.saspatlas.com/)[16].

One of the proteins identified as differentially present in the medium fraction was the N-terminal extracellular domain of Ptk7 (Fig. 2b–e, Supplementary Data 4). Ptk7 transcripts are robustly expressed in MEFs, and can also be detected in intestinal organoids (Supplementary Fig. 2c). The N-terminal domain of Ptk7 is known to be shed after cleavage by either Matrix Metalloprotease 14 (MMP14) or A

Disintegrin And Metalloprotease 17 (ADAM17) (Fig. 2d)[58,59,71], and MMP − mediated shedding of N-terminal Otk influences ISC migration in *Drosophila*[68]. Consistently, peptides detected in SCM mapped to the N-terminal extracellular domain (Fig. 2e). We confirmed the selective presence of the soluble/shed form of Ptk7 (sPtk7) in the medium fraction of SCM using a commercially available antibody against the N-terminal domain of Ptk7 (Supplementary Fig. 2d). Conditioned media (CM) from both doxorubicin- and irradiation-induced senescent

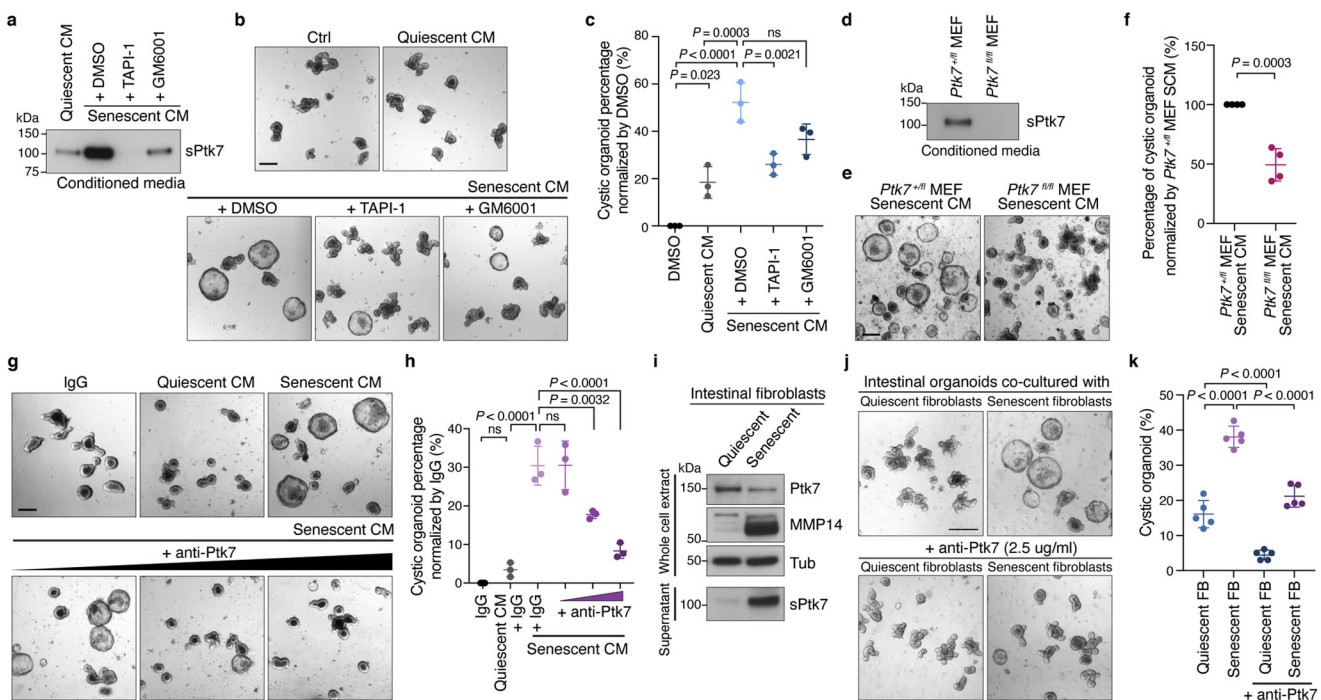

**Fig. 3 | SCM-induced cystic organoid phenotype is dependent on Ptk7.**
**a** Western blot for sPtk7 in conditioned media collected from quiescent and senescent MEFs treated with vehicle or inhibitors for 24 h. Conditioned media corresponding to the same number of cells was loaded. **b** Representative images of organoids cultured in corresponding conditioned media. Scale bar, 200 μm. **c** Quantification of data from (**b**). The percentage of cystic organoids is shown. **d** Western blot for sPtk7 in conditioned media collected from senescent Ptk7[+/fl] and Ptk7[fl/fl] fibroblasts. Conditioned media corresponding to the same number of cells was loaded. **e** Representative images of organoids cultured in corresponding conditioned media. Scale bar, 200 μm. **f** Quantification of data from (**e**). **g** Representative images of organoids cultured in conditioned media together with

IgG (5 μg/ml) or increasing concentration of anti-Ptk7 antibodies (1, 2, and 5 μg/ml). Scale bar, 200 μm. **h** Quantification of data from (**g**). **i** Western blots of full-length and secreted Ptk7 and MMP14 in whole-cell extract of quiescent and senescent primary intestinal fibroblasts or conditioned media collected from the fibroblasts. Tubulin was used as loading control. **j** Representative images of organoids co-cultured with quiescent or senescent intestinal fibroblasts in the presence of IgG or anti-Ptk7. Scale bar, 200 μm. **k** Quantification of data from (**j**). In (**c**), (**f**), (**h**), and (**k**), data are presented as means ± s.d. One-way ANOVA followed by Tukey's post-hoc multiple comparison tests. Each dot represents a result from organoids established from a mouse, $n = 3$ in (**c**) and (**h**), $n = 4$ in (**f**), and $n = 5$ in (**k**). See also Fig. S2. Source data are provided as a Source Data file.

cells contained higher levels of sPtk7 compared to CM from quiescent cells (Fig. 2f; loading was normalized by the number of cells cultured in the corresponding media), and increased sPtk7 was detected in the SASP Atlas of senescent human lung fibroblasts (Supplementary Data 3). These observations indicate that secreted Ptk7 is a common component of the SASP.

### Increased shedding of sPtk7 expresses in the aging mouse intestine

Since senescent cells increase with age in many tissues[1,2,8], we asked whether secreted Ptk7 could be detected at a higher frequency in intestines of old mice. Ptk7 transcripts and proteins were detected in both epithelial and stromal cells of the mouse intestine, with higher expression in stromal cells (Supplementary Fig. 2e and Fig. 2g, h). We did not detect any differences in Ptk7 transcript levels between young and old intestine (Supplementary Fig. 2f). Consistently, total Ptk7 protein levels did not change with age. However, Ptk7 shedding was strongly increased in the old intestine, as indicated by the increased levels of shed N-terminal Ptk7 (Fig. 2i, j).

ADAM and MMP expression increase in senescent cells[72–75], which may explain the increased release of sPtk7 into the SCM. To test this possibility, we exposed senescent MEFs to metalloprotease inhibitors previously shown to block Ptk7 shedding[59] and collected SCM. Metalloprotease inhibition reduced the levels of sPtk7 in SCM (Fig. 3a). Organoids exposed to this SCM (after dialysis to remove metalloprotease inhibitors) showed reduced cystic phenotypes (Fig. 3b, c). To validate that sPtk7 is responsible for the cystic organoid phenotype, we exposed organoids to SCM collected from *Ptk7*-deficient fibroblasts.

SCM collected from *Ptk7*-deficient fibroblasts contained reduced levels of sPtk7 (Fig. 3d), and organoids exposed to this SCM showed reduced cystic organoid phenotypes compared to its counterpart (Fig. 3e, f). We further blocked Ptk7 directly in SCM by adding anti-Ptk7 antibodies targeting the N-terminal domain to the organoid culture, which suppressed the cystic organoid phenotype in a dose-dependent manner (Fig. 3g, h).

We asked whether co-culturing organoids with intestinal fibroblasts recapitulates what we observed using conditioned media. Cellular quiescence and senescence of intestinal fibroblasts were induced by serum deprivation and doxorubicin treatment, respectively. SA-β-gal staining confirmed senescence in the intestinal fibroblasts (Supplementary Fig. 2g). Senescent intestinal fibroblasts showed increased Ptk7 shedding, which coincides with increased levels of MMP14 (Supplementary Fig. 2h and Fig. 3i). While organoids co-cultured with quiescent intestinal fibroblasts displayed normal budding organoid morphology, organoids co-cultured with senescent intestinal fibroblasts exhibited cystic morphology that resembled SCM exposed organoids (Fig. 3j, k). Inhibition of Ptk7 suppressed this cystic phenotype caused by co-culture with senescent intestinal fibroblasts (Fig. 3j, k).

Together, our results identify sPtk7 as a SASP factor that is released from senescent cells by metalloproteases and disrupts intestinal organoid morphology.

### sPtk7 acts through the noncanonical Wnt signaling

How does Ptk7 cause the cystic organoid phenotype? Several studies implicate Ptk7 in Wnt signaling, identifying interactions of Ptk7 with several components of canonical and noncanonical Wnt pathways in a

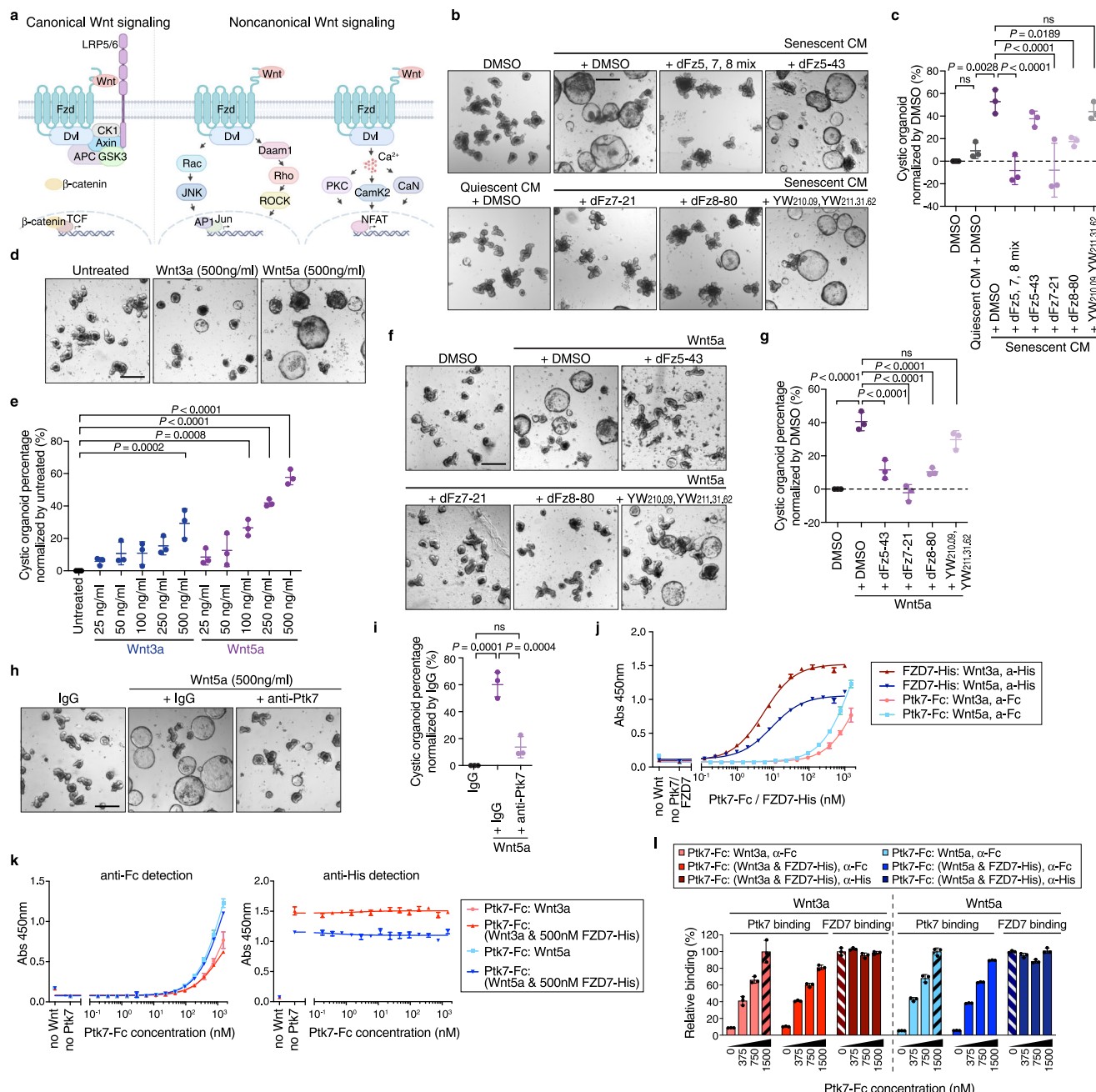

**Fig. 4 | Ptk7 acts through noncanonical Wnt signaling. a** Schematic representation of canonical and noncanonical Wnt signaling pathways. While canonical and noncanonical Wnt signaling pathways share components such as FZD and Dvl, other components such as LRP5/6 and β-Catenin are specific for canonical Wnt signaling. The figure was created with BioRender.com. **b** Representative images of organoids cultured in corresponding conditioned media with DMSO or inhibitors. Scale bar, 200 μm. **c** Quantification of data from (**b**). The percentage of cystic organoids in organoids cultured in conditioned media is shown. **d** Representative images of organoids treated with DMSO (carrier), Wnt3a or Wnt5a. Scale bar, 200 μm. **e** Quantification of data from (**d**). The percentage of cystic organoids in organoids cultured in conditioned media is shown. **f** Representative images of organoids cultured in Wnt5a with DMSO or inhibitors. Scale bar, 200 μm. **g** Quantification of data from (**f**). mean ± s.d. percentage of cystic organoids in organoids cultured in conditioned media. **h** Representative images of organoids

cultured in Wnt5a with IgG or anti-Ptk7 antibodies. Scale bar, 200 μm.
**i** Quantification of data from (**h**). mean ± s.d. The percentage of cystic organoids in organoids cultured in conditioned media is shown. In (**c**), (**e**), (**g**), and (**i**), data are presented as means ± s.d. One-way ANOVA followed by Tukey's post-hoc multiple comparison tests. Each dot represents a result from organoids established from a mouse, *n* = 3. **j** Representative ELISA binding curves showing binding of Ptk7 or FZD7 to Wnt ligands. **k** Representative ELISA binding curves showing binding of Ptk7 to Wnt ligands in the absence or presence of 500 nM FZD7. In (**j**) and (**k**), data are presented as means ± s.d. *n* = 3 (**l**) Relative Ptk7 and FZD7 binding to Wnt ligands from (**k**). Relative Ptk7 binding was calculated by normalization to Ptk7 and Wnt ligand binding at 1500 nM in the absence of FZD7 (black striped bar). Relative FZD7 binding was calculated by normalization to FZD7 binding to Wnt ligands in the absence of Ptk7 (white striped bar). Data are presented as means ± s.d. See also Fig. S3. Source data are provided as a Source Data file.

---

context-dependent manner (Fig. 4a)[62,64,65,76–79]. However, whether Ptk7 interacts directly with Wnt co-receptors, or plays a role in β-catenin-dependent or independent Wnt signaling in intestinal stem cells remains unclear.

Since exogenous Wnt3a can also cause a cystic phenotype in intestinal organoids[80], we tested for a direct effect of Wnt secretion by senescent cells. We treated senescent MEFs with a Porcupine inhibitor to block Wnt secretion[81]. Organoids exposed to SCM from these MEFs

(after dialysis to remove the inhibitor) still exhibited cystic phenotypes, confirming that Ptk7–but not Wnt ligands–in the SCM causes morphological alterations in organoids (Supplementary Fig. 3a–d). Furthermore, a Wnt reporter assay indicated no activation of β-Catenin-dependent Wnt signaling by SCM (Supplementary Fig. 3e).

To test whether sPtk7 acts through the Wnt signaling pathway, and dissect the role of different Wnt receptors in the response to sPtk7, we used inhibitory peptides. In addition to the FZD7-specific peptide inhibitor dFz7-21[82], we identified peptides binding to the cysteine-rich domains of FZD5 and FZD8 from naïve peptide libraries by phage display (dFz5-43 and dFz8-80, respectively) (Supplementary Fig. 3f, g). Surface plasmon resonance (SPR) of dFz5-43, dFz7-21, and dFz8-80 against immobilized FZD5-Fc, FZD7-Fc, and FZD8-Fc showed specificity and affinity of the peptide inhibitors to their targets (Supplementary Fig. 3h–j). dFz5-43 binds FZD5 as well as FZD8, with lower affinity to FZD5, whereas dFz8-80 selectively binds FZD8. To target the canonical FZD co-receptor Low-density lipoprotein Receptor-related Protein 6 (LRP6), we generated antibody Fabs from the previously reported antibodies YW210.09 and YW211.31.62, which bind the E1-E2 and E3-E4 domains of LRP6, respectively (Supplementary Fig. 3k,l)[83,84]. These Fabs also bound LRP5, as determined by SPR (Supplementary Fig. 3m, n).

Combined inhibition of FZD5, 7 and 8, as well as selective inhibition of FZD7 or FZD8, but not of FZD5, strongly suppressed the cystic phenotype induced by SCM, confirming a role for these FZD receptors in the SASP-induced phenotype (Fig. 4b, c). Inhibition of LRP5/6, in turn, failed to rescue the cystic phenotype, suggesting that sPtk7 acts through a FZD-dependent but LRP5/6-independent Wnt signaling pathway.

To further examine the effect of Wnt signaling on organoids, we added Wnt3a and Wnt5a recombinant proteins to crypt cultures. While both Wnt3a and Wnt5a dose-dependently increased organoids with cystic morphology, there was a noticeable difference in organoid size (Fig. 4d, e). Crypts treated with Wnt5a developed cystic organoids that were comparable to organoids exposed to SCM, but larger than those treated with Wnt3a. Wnt11, another noncanonical Wnt ligand that acts through FZD7[85,86], failed to induced cystic organoids (Supplementary Fig. 3o, p). Similar to SCM, FZD receptor inhibition reduced cystic organoids generated by Wnt5a, but inhibition of LRP5/6 did not (Fig. 4f, g).

The percentage of cystic organoids caused by Wnt5a was reduced by Ptk7 inhibition (Fig. 4h, i), suggesting an interaction between Wnt5a and Ptk7 that contributes to cystic organoid morphology. To test this possibility, we developed an ELISA to probe binding between purified recombinant proteins - Ptk7 extracellular domain (sPtk7), FZD7 CRD domain and Wnt ligands (Supplementary Fig. 3q). sPtk7 and FZD7 bind both Wnt3a and Wnt5a, but sPtk7 bound Wnt5a with higher affinity, whereas FZD7 showed the opposite (Fig. 4j). Since both sPtk7 and FZD7 bind Wnt ligands, we asked whether Ptk7 and FZD7 compete for binding to Wnt ligands. sPtk7 binding to either Wnt3a or Wnt5a was unaffected by a saturating amount of FZD7 CRD, and the interaction between FZD7 CRD and these Wnt ligands was not affected by Ptk7 (Fig. 4k, l), suggesting that Ptk7 and FZD7 bind different sites on Wnt. Together with the observation that inhibition of either FZD or Ptk7 rescued the cystic organoid phenotype caused by Wnt ligands, these results suggest that Ptk7/FZD7/Wnt form a ternary complex to activate Wnt signaling.

## Cytosolic Ca²⁺ oscillations in intestinal organoids

Wnt5a-mediated noncanonical signaling elicits cytoskeletal changes through activation of Rho and Rac GTPases and modulation of cytosolic Ca²⁺ levels[87–90]. We and others have shown that intestinal stem cells in *Drosophila* exhibit regular cytosolic and mitochondrial Ca²⁺ oscillations, and that modulation of these oscillations in response to mitogenic signals control ISC proliferation and metabolic

reprogramming[91–93]. Whether Ca²⁺ signaling also influences intestinal stem cell activity in the mammalian gut remains unclear. To assess possible effects of noncanonical Wnt signaling on Ca²⁺ levels in mammalian ISCs, we expressed the fluorescent Ca²⁺ indicator GCaMP6f under control of Lgr5-Cre and performed Ca²⁺ imaging in live intestinal organoids (Fig. 5a). In the absence of stimuli, Ca²⁺ oscillations were observed in multiple cells of organoid crypts (Fig. 5b, Supplementary Movie 1). In unchallenged organoids, baseline cytosolic Ca²⁺ levels and the frequency and intensity of oscillations were variable and did not reveal an obvious pattern (Fig. 5c, d, Supplementary Movie 2)

When exposed to Wnt5a, however, cytosolic Ca²⁺ levels and oscillations increased (Fig. 5c–e, Supplementary Fig. 4a, Supplementary Movie 3). This increase was distinct from Wnt3a treatment, which resulted in decreased cytosolic Ca²⁺ levels (Supplementary Fig. 4b), but also occurred when organoids were exposed to SCM. SCM, but not QCM, increased the frequency and magnitude of cytosolic Ca²⁺ oscillations (Fig. 5f, g; Supplementary Fig. 4c, d). Addition of SCM in the presence of anti-Ptk7 failed to induce any changes in Ca²⁺ oscillations, suggesting that sPtk7 stimulates the Wnt/Ca²⁺ pathway in ISCs (Fig. 5h, Supplementary Fig. 4e). Ptk7 inhibition in unchallenged organoids decreased cytosolic Ca²⁺ oscillations, suggesting a role for endogenous Ptk7 in modulating Ca²⁺ signaling (Fig. 5h, Supplementary Fig. 4e).

## Inhibition of Ca²⁺ signaling suppresses the cystic phenotype

To explore the mechanistic link between increased Ca²⁺ signaling and the cystic organoid phenotype, we asked whether inhibition of Ca²⁺ signaling components suppresses SCM or Wnt5a-induced cystic organoid phenotypes. Trifluoperazine (TFP), an antagonist of the multifunctional Ca²⁺ binding protein Calmodulin (CaM)[94], reduced the percentage of cystic organoids induced by SCM or Wnt5a (Fig. 5i, j, Supplementary Fig. 5a, b). Inhibition of Calcineurin by FK506 or cyclosporin A had no effect on this phenotype, but inhibition of CaMKK and CaMKII by STO-609 and KN-62, respectively, suppressed the cystic organoid phenotype (Fig. 5k, l, Supplementary Fig. 5c–e). These data suggest that cytosolic Ca²⁺ release and subsequent activation of Calmodulin / CaMK signaling lead to the cystic organoid morphology.

## YAP/TEAD target gene enrichment in SASP-exposed organoids

We performed transcriptome analysis to gain further insight into changes elicited in intestinal organoids by SCM and Ptk7. Intestinal organoids were cultured in complete media, QCM or SCM (with or without anti-Ptk7), and complete media with Wnt5a (with or without anti-Ptk7) for 2 days, and subjected to RNAseq analysis (Fig. 6a; all cultures were exposed to either anti-Ptk7 or IgG control). Principal component analysis (PCA) of 36 organoid transcriptomes revealed a marked similarity between the transcriptomes of organoids cultured in complete media and those in QCM, but significant differences between organoids cultured in other conditions (Fig. 6b). Differential gene expression analysis revealed that close to half the genes induced by Wnt5a were also induced by SCM, while the majority of SCM-induced genes were also induced by Wnt5a, consistent with a major contribution of noncanonical Wnt signaling to the SCM-induced cystic organoid phenotype. Of genes differentially regulated by Wnt5a and SCM, about half were Ptk7 dependent, confirming a critical role for Ptk7 in the SCM-induced phenotype (Fig. 6c).

Transcription factor binding motif analyses using i-cisTarget[95,96] revealed a clear difference between Ptk7-dependent and -independent differentially expressed genes (DEGs). Ptk7-dependent DEGs displayed strong enrichment for TEAD binding motifs, whereas Ptk7 independent DEGs showed enrichment for Hdac2, Hnf4g and Hnf1a binding motifs (Fig. 6d). Most Ptk7-dependent DEGs with TEAD binding motifs increased expression in response to SCM or Wnt5a, and are known YAP target genes (Fig. 6e, Supplementary Fig. 6a–d).

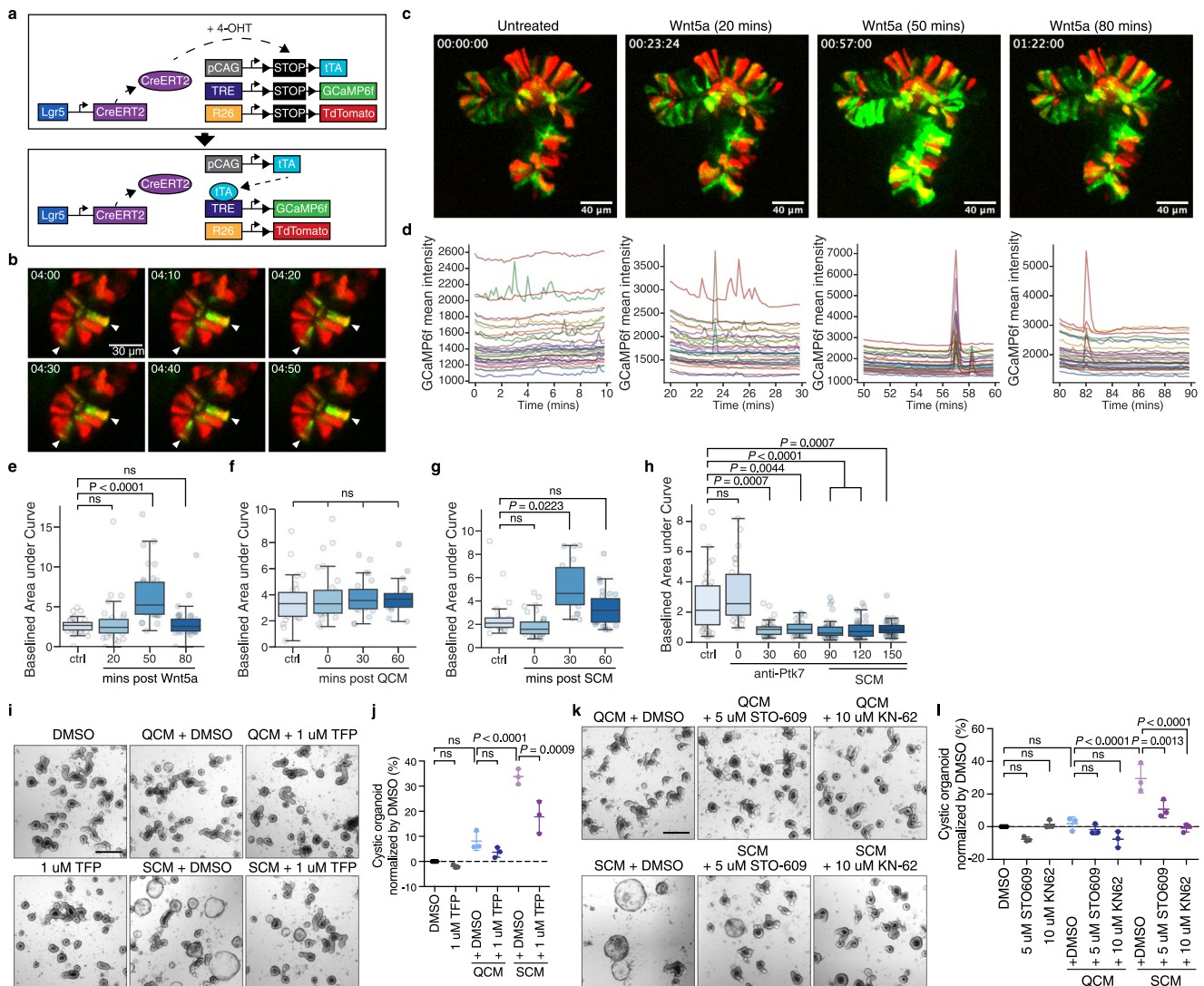

**Fig. 5 | Ptk7 modulates cytosolic Ca²⁺ oscillations in intestinal organoids. a** Lgr5-CreERT2-mediated expression of GCaMP6f in this experiment. Cre removes stop cassettes upon addition of 4-OHT, allowing expression of tTA, GCaMP6f, and tdTomato. **b** Time lapse images showing Ca²⁺ flashes in the crypt domain of a mouse intestinal organoid. GCaMP6f and tdTomato expression is shown in green and red, respectively. Scale bar, 30 µm. **c** Representative frames from live imaging of an organoid expressing GCaMP6f before and after Wnt5a treatment. $n = 5$ **d** GCaMP3f mean intensity traces of 10-minute live imaging matching the frames shown in (**c**). Each trace represents one cell. Boxplots showing baselined area under curve of live imaging before and after Wnt5a (**e**), QCM (**f**), SCM (**g**), or SCM and anti-Ptk7 antibody (**h**) treatment. The graphs show results from a representative experiment. $n = 5$ in (**e**), $n = 3$ in (**f**), $n = 6$ in (**g**), and $n = 6$ in (**h**). In (**h**), organoids were imaged every 30 min. 90 min after the addition of the antibodies, SCM was

added to the media containing anti-Ptk7. The graphs show results from a representative experiment. $n = 6$. In (**e–h**), boxplots display median ± interquartile range with whiskers marking 1.5 times the interquartile range. Kruskal–Wallis test followed by Dunn's post-hoc multiple comparison tests. **i** Representative images of organoids cultured in QCM or SCM with DMSO or 1 µM TFP. Scale bar: 200 µm. **j** Quantification of data from (**i**). mean ± s.d. percentage of cystic organoids in organoids cultured in conditioned media. **k** Representative images of organoids cultured in QCM or SCM with DMSO, 5 µM STO-609, or 10 µM KN-62. **l** Quantification of data from (**k**). mean ± s.d.; $n = 3$; two-tailed $t$-test. The percentage of cystic organoids in organoids cultured in conditioned media is shown. In (**j**) and (**l**), One-way ANOVA followed by Tukey's post-hoc multiple comparison tests. Each dot represents a result from organoids established from a mouse, $n = 3$. See also Figs. S4, S5. Source data are provided as a Source Data file.

A role for YAP in SCM-mediated effects in organoids was supported by our snRNA-seq analysis. While we did not see any clear changes in ISC/CBC specific genes, we observed an overall increase of YAP target gene expression across different clusters (Supplementary Fig. 6e–h). Increase of YAP target gene score compared to ISC/CBC signature was particularly noticeable in the fetal signature gene expressing cell cluster (Supplementary Fig. 6i, j). YAP activation promotes expression of genes constituting a fetal signature in intestinal organoids and in the injured intestinal epithelium[53,97–99]. Consistently, we observed expression of several fetal signature genes in cells with high YAP target gene expression (Supplementary Fig. 6k).

Altogether, these data suggest that the cystic organoid phenotype caused by Ptk7 or Wnt5a may result from elevated YAP/TEAD activity. This hypothesis was supported by the relative nuclear localization of YAP in SCM or QCM exposed organoids. Active YAP translocates to the nucleus to form a complex with TEADs[48]. Whole mount immunostaining showed that YAP localizes mainly to the cytosol in budding organoids in QCM (Fig. 7a, top). However, cystic organoids in SCM or after Wnt5a treatment displayed increased nuclear localization of YAP (Fig. 7a, middle and bottom). Interestingly, organoid morphology correlated strongly with YAP localization. In rare cystic organoids in QCM, YAP also localized to the nucleus, while the few normally

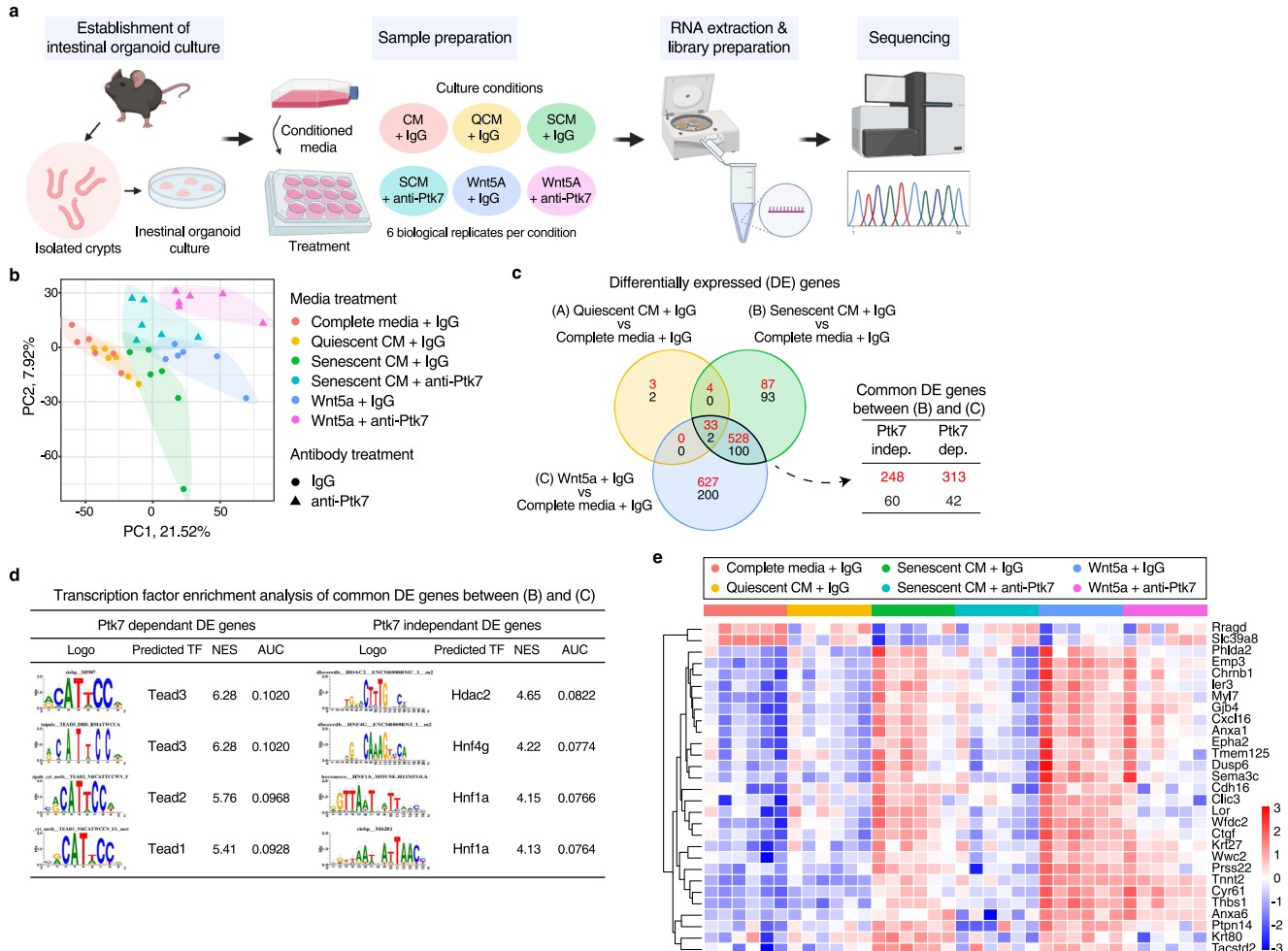

**Fig. 6 | Transcriptomes of organoids shows enrichment of TEAD binding motifs among Ptk7-dependent DE genes. a** A diagram showing the workflow of intestinal organoid culture, media treatments, sample preparation, and RNAseq. The figure was created with BioRender.com. **b** Principal component analysis (PCA) of RNAseq results from the indicated organoids. Each point represents individual samples, and sample groups are indicated by using different colors as indicated in the legend provided. **c** Venn diagram showing overlap in differentially expressed genes (DEGs) between the indicated conditions. Numbers of DEGs upregulated and down-regulated are shown in red and black, respectively. FDR < 0.05 and fold change >1.5 were used to acquire the list of DEGs. Overlapping DEGs from SCM and Wnt5a treated organoids were further divided into two groups by Ptk7 dependency. **d** Transcription factor enrichment analysis of common DEGs from (**c**) using i-cisTarget. Highly enriched motifs and predicted transcription factors for Ptk7-dependent and independent DEGs were ranked by normalized enrichment score (NES). **e** Heatmap showing expression of selected Ptk7-dependent DEGs with TEADs binding motifs in different conditions. Heatmaps of Ptk7-dependent DEGs with different TEAD binding motifs (TEAD 1–4) are found in Supplementary Fig. 6a–d. See also Fig. S6.

budding organoids after SCM or Wnt5a treatment exhibited cytosolic YAP (Supplementary Fig. 7a, b). Additionally, the morphological changes observed after switching CM corresponded to the nuclear localization of YAP (Supplementary Fig. 7c). This finding is consistent with the reported need for heterogeneous YAP activation for symmetry breaking in organoids[21], and suggests that constitutive YAP activation and YAP/TEAD target gene expression may cause the cystic organoid phenotype.

### Inhibition of YAP reverses cystic phenotype
To test this hypothesis, we asked whether inhibition of nuclear YAP rescues the cystic organoid phenotype. Verteporfin is a small molecule that inhibits YAP[100]. Verteporfin strongly reduced the cystic organoid percentage due to QCM and SCM (Supplementary Fig. 7d, Fig. 7b, c). We also used an irreversible covalent TEAD inhibitor, TED347, which blocks TEAD binding to YAP[101]. Similar to verteporfin, TED347 reduced the percentage of cystic organoids (Fig. 7d, e).

$Ca^{2+}$ signaling has been implicated in the regulation of YAP, but the effect of perturbing cytosolic $[Ca^{2+}]$ on YAP nuclear translocation may be context-dependent[102]. To test whether $Ca^{2+}$ signaling regulates

YAP localization downstream of SCM/sPtk7/Wnt5a in organoids, we inhibited CaM, CaMKK and CaMKII using TFP, STO-609 and KN-62, respectively. Inhibition of these $Ca^{2+}$ signaling components reversed SCM- or Wnt5a-induced nuclear translocation of YAP (Fig. 7f), as well as the cystic organoid phenotype (Fig. 5j–m). Together, our results demonstrate that noncanonical Wnt-$Ca^{2+}$ signaling promotes YAP/TEAD activation in intestinal stem cells, resulting in the cystic morphologies adopted by organoids exposed to SCM or Wnt5a.

### Discussion
The aging intestinal epithelium displays reduced regenerative capacity. Several age-dependent changes such as reduced Wnt signaling, elevated Cdc42 activity, and metabolic changes have been suggested to cause reduced intestinal stem cell activity in the intestinal crypt[36,37,103–106]. At the same time, aging is associated with dramatically increased colorectal cancer incidence, likely resulting from intestinal stem cell deregulation. Whether and how senescent cells contribute to these changes in intestinal homeostasis remains unclear.

Our data suggest a possible role for the SASP in the age-related deregulation of ISC function and differentiation (Supplementary

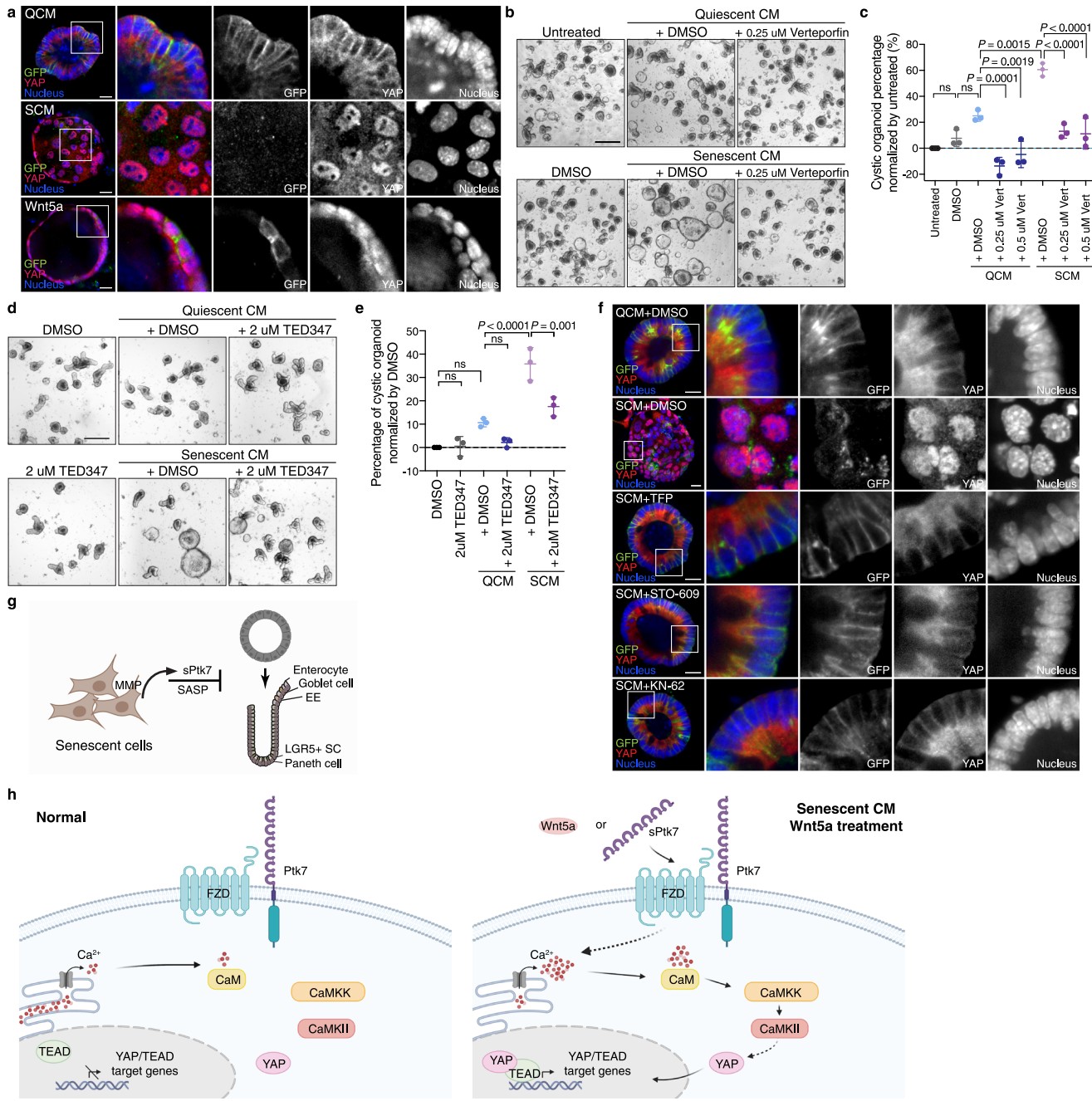

**Fig. 7 | Correlation between nuclear YAP localization and cystic organoid morphology and connection to Ca²⁺ signaling. a** Whole mount immunostaining of organoids for YAP. Organoids established from Lgr5-DTR-GFP mice were stained for GFP, YAP, and nucleus in green, red, and blue respectively. Representative images of an organoid from each condition are shown. $n = 8$ per each condition. Enlarged images of boxed area are shown at the right. Scale bar: 20 μm. **b** Representative images of organoids cultured in QCM or SCM with DMSO or verteporfin. Scale bar: 200 μm. **c** Quantification of data from (**b**). The percentage of cystic organoids in organoids cultured in conditioned media is presented as means ± s.d. **d** Representative images of organoids cultured in QCM or SCM with DMSO or TEAD inhibitors. Scale bar: 200 μm. **e** Quantification of data from (**d**). The percentage of cystic organoids in organoids cultured in conditioned media is presented as means ± s.d. In (**c**) and (**e**), each dot represents a result from organoids established from a mouse, $n = 3$. One-way ANOVA followed by Tukey's post-hoc

multiple comparison tests. **f** Whole mount immunostaining of organoids for YAP in organoids treated with QCM or SCM with DMSO, TFP, STO-609, or KN-62. Scale bar: 20 μm. Enlarged images of boxed area are shown on the right. **g** Model for the disruption of organoid differentiation by sPtk7 secreted by senescent cells. Senescent cells shed sPtk7 through the action of Matrix metalloproteinases (MMPs), disrupting crypt formation and epithelial cell differentiation in intestinal organoids. **h** In contrast to normal condition, the presence of sPtk7 or Wnt5a stimulates Ca²⁺ signaling in intestinal epithelial cells. Activation of the CaM/CaMKK/CaMKII pathway promotes translocation of YAP to the nucleus where it binds to TEADs and upregulates YAP/TEAD target genes. These changes in gene expression affect stem cell differentiation, resulting in cystic organoid morphology. **g** and **h** were created with BioRender.com. See also Fig. S7. Source data are provided as a Source Data file.

Fig. 6l, m). The SASP (including factors like sPtk7, which are not technically secreted but are shed as a consequence of senescent cell surface remodeling) is believed to be a critical part of the contribution of senescent cells to age-related disease, primarily by influencing the tissue microenvironment and spreading senescence through a "bystander effect"[107–111]. Accordingly, selective elimination of senescent cells improves many aging symptoms and disease phenotypes[2,3,8,9,112]. Many SASP components have been identified using proteomic and transcriptomic methods[13,16,17,109,113–115]. However, how individual SASP factors influence the tissue microenvironment and contribute to tissue dysfunction is incompletely understood. Our study identifies sPtk7 as a critical SASP factor that has a direct and reversible impact on intestinal stem cell proliferation and differentiation (Fig. 7g, h).

Ptk7 has previously been identified as a surface marker of human colonic stem cells, with cells expressing high levels of Ptk7 displaying the highest self-renewal and re-seeding capacity in 3D organoid culture[116]. Our data show that Ptk7 is also expressed in fibroblasts and epithelial cells of the mouse small intestine, and that shedding of the N-terminal domain of Ptk7 is increased in the gut of old mice. Of note, we did not detect sPtk7 in the serum of mice at any age, suggesting that sPtk7 acts locally rather than systemically. Accordingly, we have found in a recent study that the N-terminal domain of the Ptk7 orthologue Otk is shed from enteroendocrine cells after intestinal epithelial injury, and acts locally to attract migrating stem cells to the site of damage[68].

Our co-culture experiments of intestinal organoids with senescent intestinal fibroblasts further show that fibroblast-derived Ptk7 impacts differentiation of intestinal stem cells, causing a cystic organoid phenotype. How this effect on intestinal stem cells influences epithelial homeostasis and regeneration remains to be established.

Previous studies have observed cystic intestinal organoid morphology after activation of β-catenin-dependent canonical Wnt signaling[117–120]. Our data indicate a distinct difference between Wnt3a and Wnt5a. Wnt5a can induce the cystic organoid phenotype by activating noncanonical Wnt/$Ca^{2+}$ signaling. sPtk7 phenocopies these Wnt5a-mediated effects, acting through $Ca^{2+}$ and YAP/TEAD signaling. Mechanistically, our data provide evidence for a possible ternary complex between sPtk7, Wnt5a and FZD7, where Wnt5a mediates the interaction between FZD7 and cell surface Ptk7. The ternary complex is required for cystic organoid formation, as inhibition of either FZD or Ptk7 suppresses the Wnt5a-induced cystic organoid phenotype. Wnt5a is known to play a role in intestinal development and wound healing[121–123]. Additional studies to explore if Ptk7 similarly influences intestinal development and wound healing will be of interest in the future.

At the molecular level, it is conceivable that sPtk7 acts similarly to Wnt5a, a notion that is supported by the fact that the fly orthologue of Ptk7 can form homo-oligomers and bind to FZD1/2 via its N-terminal domain[79]. It is also possible that a yet to be identified cell surface co-receptor inhibits ternary Wnt/FZD7/Ptk7 complex formation by sequestering cell surface Ptk7, and that sPtk7 binds this co-receptor to liberate cell surface Ptk7 and enable or promote complex formation with Wnt/FZD7 (especially at low endogenous Wnt concentrations). In such a scenario, recombinant Wnt5a (high Wnt) might force ternary complex formation without the need for secreted Ptk7.

Our observation of cytosolic $[Ca^{2+}]$ oscillations in mammalian intestinal epithelial cells is reminiscent of $[Ca^{2+}]$ oscillations found to regulate stem cell quiescence in the fly intestinal epithelium[91]. Supporting the proposed role of sPtk7 in modulating Wnt5a-mediated signaling, we find that sPtk7 and Wnt5a modulate cytosolic $Ca^{2+}$ oscillations, while Wnt3a does not. Interestingly, inhibition of Ptk7 strongly suppressed baseline $Ca^{2+}$ oscillations, suggesting a role for Ptk7 in maintaining homeostatic $Ca^{2+}$ signaling. Further studies are needed to fully explore the role of Ptk7 in $Ca^{2+}$ signaling in stem cells during homeostasis and regeneration.

In intestinal organoids, our data indicate a critical role for $Ca^{2+}$/CaM/CaMKK/CaMKII signaling in promoting constitutive nuclear translocation of YAP and YAP/TEAD target gene expression after Wnt5a/sPtk7 exposure. It remains to be established, however, whether this influence of $Ca^{2+}$ signaling on YAP nuclear translocation is direct or whether it is a consequence of more general cellular changes triggered by elevated cytosolic $[Ca^{2+}]$. The constitutive activation of YAP/TEAD target expression causes the cystic phenotype, consistent with reports from early organoid development in which abnormal YAP activation prevents symmetry breaking and results in spheroid organoids[21].

We propose an impact of YAP activation on epithelial cell differentiation, as exposure to SCM resulted in reduced numbers of differentiated cell types, especially of cells in the secretory lineage, as well as the induction of a fetal gene signature. Accordingly, loss of differentiated cell types and induction of fetal genes has been reported when YAP is activated in the mouse intestine[51,52,54]. This activation of YAP is important for tissue regeneration, and we propose that the observed sPtk7-mediated YAP activation also regulates epithelial regeneration in young animals. Such a role for sPtk7 would be consistent with the recently described function of the *Drosophila* orthologue of Ptk7, Otk[68]. After tissue injury, the N-terminal domain of Otk is shed from enteroendocrine cells and promotes noncanonical Wnt signaling in nearby intestinal stem cells, promoting their migration to the site of injury. Our data suggest that such an evolutionarily conserved role for transient release of sPtk7 during regeneration results in tissue dysfunction in aging animals, where chronic exposure to elevated sPtk7 (as observed in our study) would prevent proper cell differentiation of the epithelium. Future studies exploring the effects of sPtk7 during tissue regeneration in the young and old mouse intestine will be of interest.

The specific loss of Olfm4+ SCs over Lgr5+ SCs we observed is interesting in this context. While both Lgr5 and Olfm4 are stem cell markers, Lgr5 is a downstream target of canonical Wnt signaling, whereas the expression of Olfm4 is induced by Notch signaling in intestinal stem cells[124]. While this would indicate a repression of Notch activity in SCM exposed crypts, the reduced differentiation of secretory cells is reminiscent of activated Notch signaling[125–127]. Clearly, the interaction between Ptk7/FZD/$Ca^{2+}$/YAP signaling and Notch signaling in intestinal stem cells and their daughter cells is an interesting topic for further study. It will be important to evaluate whether and how such an interaction causes differentiation defects or skewed cell fate decisions in the ISC lineage.

The SASP has been postulated as a main culprit for tissue dysfunction in aging and disease, in part because proinflammatory factors that are part of the SASP can cause excessive immune cell recruitment/activation and consequential tissue damage. Notably, however, disruption of normal differentiation in the ISC lineage by SASP factors in our model occurs in the absence of immune cells. Our data thus reveal that factors secreted by senescent cells can also have direct effects on epithelial regeneration and stem cell function, effects that potentially contribute to age-related tissue dysfunction and oncogenic transformation.

## Methods

### Mice

All mice were used in accordance with protocols approved by Genentech's Institutional Animal Care and Use Committee and adhere to the NRC Guidelines for the Care and Use of Laboratory Animals. C57BL/6 (JAX:000664), GCaMP6f (JAX:030328)[128], and ROSA26-tdTomato (JAX:007914)[129] mice were purchased from the Jackson Laboratory. Three- to four-month-old male C57BL/6 mice (CRL:027) and 16–18 months old male C57BL/6 mice (CRL:701) were purchased from Charles River laboratories and further aged until 28–30 months of age. Ptk7$^{fl/fl}$ (Gene ID: 71461) mice were purchased from genOway. Lgr5$^{DTR/EGFP}$ and Lgr5$^{CreERT2}$ mice were generated previously

previously[130], and GCaMP6f; ROSA26-tdTomato mice were crossed with Lgr5[CreERT2] mice to enable expression in Lgr5+ intestinal stem cells and their progeny.

## Mouse embryonic fibroblast culture

Mouse embryonic fibroblasts (MEFs) were purchased from Sigma (#PMEF-CFL) or isolated from Ptk7[+/fl] and Ptk7[fl/fl] embryos using the Mouse embryonic Fibroblast Isolation Kit (Thermo Fisher Scientific, #88279). For Ptk7[+/fl] and Ptk7[fl/fl] MEFs, TAT-CRE Recombinase (Sigma-Aldrich, #SCR508) was used to induce deletion of *Ptk7*. MEFs were cultured in Dulbecco's Modified Eagle Medium (DMEM; #D6429) supplemented with 10% fetal bovine serum (FBS; Thermo Fisher Scientific, #2614079) and 1% penicillin and streptomycin (5000 U/mL and 5000 µg/mL each; Thermo Fisher Scientific, #15070063) at 37 °C and 5% $CO_2$. Once cellular quiescence or senescence was induced, MEFs were maintained at 37 °C, 5% $CO_2$, and 3% $O_2$ until conditioned media were collected.

## Induction of quiescence and senescence

MEFs were split a day before induction of quiescence or senescence. To induce quiescence, MEFs were cultured in low serum (DMEM supplemented with 0.5% FBS and P/S at 37 °C, 5% $CO_2$, and 3% $O_2$ for 2 days. Cellular senescence was induced by X-ray irradiation (total 10 Gy) or 250 nM doxorubicin treatment for 3 days (doxorubicin; Sigma-Aldrich, #44583). After X-ray irradiation and change to fresh culture media (DMEM supplemented with 10% FBS and 1% P/S), MEFs were cultured at 37 °C, 5% $CO_2$, and 3% $O_2$ for 9 days. For doxorubicin treatment, cells were maintained at 37 °C, 5% $CO_2$ during the 3-day doxorubicin treatment. Then, MEFs were washed with serum-free DMEM twice and cultured in DMEM supplemented with 10% FBS and 1% P/S at 37 °C, 5% $CO_2$, and 3% $O_2$ for 6 days. Culture media was changed every 2–3 days.

## Collection of conditioned media

Twenty-four hours before collecting conditioned media (CM), MEFs were washed with serum-free DMEM twice, and serum-free DMEM was added. After 24 h in serum-free DMEM, this CM was collected, filtered using Steriflip filters (Millipore, #SE1M179M6) to remove cell debris, and concentrated using centrifugal filter units (Amicon ultra-15 centricon; Millipore, # UFC900324). To collect CM without sPtk7, TAPI-1 (Millipore, #59053) or GM6001 (Sigma-Aldrich, #364206) containing serum-free DMEM instead of serum-free DMEM was added to the cells. To inhibit Wnt secretion, IWP-L6 (Sigma-Aldrich, #504819) containing serum-free DMEM instead of serum-free DMEM was added to the cells. CM was filtered and dialyzed for 24–48 h to remove the inhibitors before concentration. Concentrated CM from equal numbers of MEFs was mixed with IntestiCult Organoid Growth Medium (Stem Cell Technologies, #06005) and added to the organoid cultures.

## SA-β-Gal staining

Staining was performed with the Senescence Detection Kit (Abcam, #ab65351) following the manufacturer's protocol. Briefly, MEFs were fixed for 15 min with 1× fixation solution, washed with PBS, then stained overnight at 37 °C in 1× staining solution. Brightfield images were captured on the Zeiss Axio Imager M2 microscope.

## Mouse intestinal organoid culture

The jejunum part of the small intestine was removed, flushed with PBS, opened longitudinally, then cut into 0.5 cm pieces. The collected tissues were washed with ice cold PBS multiple times by vortexing and incubated in PBS containing 2.5 mM EDTA for 5 min at 37 °C. After gentle shaking and aspiration, tissues were incubated in PBS containing 5 mM EDTA for 7 min at 37 °C. Tissues were then washed with PBS to remove EDTA and vigorously shaken to release crypts. The released crypts were passed through 70 µm strainers, washed with PBS, and pelleted by centrifugation at $200 \times g$. Resuspended crypts were mixed with Matrigel (Corning, #356231; Lot#8218007) and pipetted into culture plates. After mixture of crypts and Matrigel solidified, Intesti-Cult Organoid Growth Medium (Stem Cell Technologies, #06005; Lots #1000021927, #1000034230, #1000037372) was added. Seven days after seeding, organoids were resuspended in dissociation buffer (Gentle Cell Dissociation Reagent; Stem Cell Technologies, #07174), dissociated by pipetting, and seeded for experiments.

## Brightfield imaging of organoids and quantification

3-4 biological replicates of organoids from different mice were used per condition. Organoids were cultured in the indicated media, inhibitors, antibodies and/or recombinant proteins for 5 d and fixed in 4% formaldehyde in PBS. Organoids were imaged with a 4× Plan Fluor objective (NA: 0.13, Nikon) on a Nikon Ti-E inverted microscope equipped with a Neo scMOS camera (Andor, Oxford Instruments), a linear encoded automated stage (Applied Scientific Instrumentation), all run by NIS Elements software (Nikon). Organoids were imaged in brightfield and images covering the whole well were stitched and focused into one image projection with an extended depth of focus module (EDF; Nikon). One hundred to 200 organoids per replicate per condition were counted and used for quantification. The number of budding organoids in each condition was normalized by the number of budding organoids in untreated or DMSO treated controls to calculate cystic organoid percentage:

$$\text{Cystic organoid percentage} = 100 \times \left(1 - \frac{Number\ of\ budding\ organoids\ from\ experiment}{Number\ of\ budding\ organoids\ from\ control}\right) \quad (1)$$

## Induction of quiescence and senescence of mouse intestinal fibroblasts and co-culture with intestinal organoids

Mouse intestinal fibroblasts were purchased from Cell Biologics (#C57-6231) and cultured in Complete Fibroblast Medium (Cell Biologics, M2267) at 37 °C and 5% $CO_2$. Cellular quiescence and senescence in mouse intestinal fibroblasts were induced by the similar methods described earlier for MEFs with minor modifications. Briefly, cellular quiescence was induced by culturing the fibroblasts in low serum (0.5% FBS), and cellular senescence was induced by 500 nM doxorubicin treatment for 5 days (doxorubicin; Sigma-Aldrich, #44583). Once cellular quiescence or senescence was induced, MEFs were maintained at 37 °C, 5% $CO_2$, and 3% $O_2$ for another 5 days. The quiescent or senescent fibroblasts were dissociated using 0.05% Trypsin-EDTA (Gibco, #25300-054) and counted for seeding. The same number of quiescent and senescent fibroblasts ($5.2 \times 10^4$ cells) were mixed with intestinal organoids making total volume of 20 µl of Matrigel: organoid culture media mix, which was mounted on a 48 well plate.

## Colony forming efficiency assay

Two millimolar EDTA in PBS was added to organoids to break Matrigel. Collected organoids were pelleted by centrifugation, washed with PBS, pelleted again, and incubated in prewarmed TrypLE Express (Gibco, #12604-013) for 8 min. The digestion was stopped by adding serum containing DMEM/F12, and the number of cells was assessed. $1 \times 10^4$ cells were seeded in total volume of 20 µl of Matrigel: organoid culture media mix per well in a 48 well plate. Once Matrigel was solidified, organoid culture media was added.

## H&E and immunofluorescence stains

Organoids were paraformaldehyde-fixed and paraffin-embedded using standard procedures. The H&E Stain Kit (Abcam, #ab245880) was used following the manufacturer's manual. For Ki67 and pH3 antibody stains, sections were incubated for 15 min in citrate buffer, blocked in normal donkey serum, and incubated with primary antibodies overnight. Sections were then incubated with fluorescent

secondary antibodies and applied with ProLong Diamond Antifade Mountant (Invitrogen, # P36970). Images were acquired with a Zeiss LSM 710. Information on antibodies is listed separately.

## Whole mount imaging of organoids and quantification of different cell types

For whole mount staining, freshly passaged crypts were mixed with Matrigel (Corning, #356231) and mounted to μ-Slide 8 well chamber slides (ibidi, #80826). Media were changed every 2–3 d. 5 days later, organoids in the chamber slides were then fixed in 4% formaldehyde in PBS for 30 min, permeabilized with 0.5% Triton X-100 for 1 h, and incubated in blocking solution (3% normal donkey serum (Sigma-Aldrich, #566460) and 5% BSA (Sigma-Aldrich, #A7979) in 0.1% Triton X-100 + PBS) for 1 h. Primary and secondary antibodies were diluted in blocking solution and added to fixed organoids. Organoids were incubated with primary antibodies overnight at 4 °C and with secondary antibodies for 2 h. ProLong Diamond Antifade Mountant (Invitrogen, # P36970) was added. Images were acquired with a Yokogawa CSU-W1/Zeiss 3i Marianas spinning disk confocal microscopy system. For each organoid, Z-planes up to 100 μm with 1 or 1.5 μm z-steps were acquired to quantify the number of cell types.

## Quantitative PCR (qPCR)

RNA was isolated using the RNeasy Plus Mini Kit (Qiagen, #74134) following the manufacturer's instructions. cDNA was synthesized using iScript cDNA Synthesis Kit (Bio-Rad, #1708890) and used in standard qPCR reactions utilizing TaqMan assays (Thermo Fisher Scientific, probes listed separately) for target genes and TaqMan Universal PCR Master Mix (Thermo Fisher Scientific, #4326708). qPCR reactions were run on the QuantStudio 6 Real-Time PCR System (Applied Biosystems, Thermo Fisher Scientific) at the following thermal cycling conditions: 30 min at 48 °C followed by 10 min at 95 °C and 40 cycles of 10 s at 95 °C and 1 min at 60 °C. Data were normalized to the housekeeping gene GAPDH.

## Immunofluorescence staining and in situ hybridization of the mouse intestine

Mouse large intestine tissues were harvested and fixed in 10% neutral buffered formalin for 24 h. Fixed tissues were then washed and processed for paraffin embedding. Formalin-fixed paraffin-embedded tissues were sectioned at 5 μm thickness and used for immunofluorescence staining and in situ hybridization. For immunofluorescence staining, sections were deparaffinized and processed using Target retrieval solution (Dako, #S1700). The sections were then blocked in 3% normal donkey serum, incubated with primary and secondary antibodies, and mounted using ProLong Diamond Antifade Mountant (Invitrogen, # P36970). In situ hybridization was performed using RNAscope mouse Ptk7 probe (ACD, #429191) and reagents (RNAscope 2.5 HD Reagent Kit; ACD, #322350) following the manufacturer's instructions.

## Proteomic sample preparation

MEFs were cultured as above, with the exception that CM was prepared by incubating MEFs with DMEM without phenol red (Gibco, #31053-028) for 24 h. Filtered CM was snap-frozen and stored at −80 °C after addition of complete protease inhibitors (Roche, #11836170001). Samples thawed on ice were additionally filtered using 70 kDa cutoff centrifugal filters (Millipore, #UFC700308). The filtrate was concentrated using 3 kDa cutoff Centricon-Plus 70 centrifugal filters to 60 ml. Additional complete protease inhibitors were added and then the concentrate was dialyzed against three changes of 4 L of 20 mM Tris, 100 mM NaCl, 5 mM EDTA, pH 8.0 at 4 °C. Samples were then loaded onto a HiTrap Q HP column (Millipore, #GE29-0513-25) equilibrated with 20 mM Tris, 5 mM EDTA, pH 8.0. The flowthrough fraction was buffer exchanged via repeat dilution with PBS and

concentration on 3 kDa cutoff filters, and loaded onto an albumin pre-coated Superdex-200 gel filtration column (Millipore, #GE28-9909-44). Sets of pooled fractions were tested for ability to promote the cystic organoid phenotype. The separated SASP fractions, containing ~2–4 μg of proteins, were concentrated from 400 μL to 25 μL using 0.5 mL centrifugal filters, after which the samples were loaded on a 1D SDS–PAGE system. The gel bands were diced and collected in tubes, and vortexed several times in dehydration buffer (25 mM ammonium bicarbonate in 50% acetonitrile and water). The samples were concentrated to dryness (speedvac), reduced with 10 mM dithiothreitol and incubated for 1 h at 56 °C with agitation. They were subsequently alkylated with 55 mM iodoacetamide and incubated for 45 min at room temperature in the dark. The diced gels pieces were washed with 25 mM ammonium bicarbonate in water, and then dehydrated once again with the dehydration buffer. The samples were then concentrated to dryness (speedvac) after which they were incubated with 250 ng trypsin for 30 min at 4 °C, then digested overnight at 37 °C with agitation. The digestions were extracted first with water and subsequently with 50% acetonitrile and 5% formic acid in water. After each step the supernatant was collected into a new tube, and pooled peptide extractions were concentrated for 2 h to dryness (speedvac), then re-suspended in 0.2% formic acid. The re-suspended samples were desalted with C18 Zip Tips, concentrated (speedvac) and re-suspended in a solution containing "Hyper Reaction Monitoring" (Biognosys) indexed retention time peptide standards (iRT) and 0.2% formic acid in water.

## Mass spectrometry

Samples were analyzed by reverse-phase HPLC-ESI-MS/MS using an Eksigent Ultra Plus nano-LC 2D HPLC system (Dublin, CA) with a cHiPLC system (Eksigent) directly connected to a quadrupole time-of-flight (QqTOF) TripleTOF 6600 mass spectrometer (SCIEX, Concord, CAN). After injection, peptide mixtures were loaded onto a C18 pre-column chip (200 μm × 0.4 mm ChromXP C18-CL chip, 3 μm, 120 Å, SCIEX) and washed at 2 μl/min for 10 min with the loading solvent ($H_2O$/0.1% formic acid). Subsequently, peptides were transferred to the 75 μm × 15 cm ChromXP C18-CL chip, 3 μm, 120 Å, (SCIEX), and eluted at a flow rate of 300 nL/min with a 3 h gradient using aqueous and acetonitrile solvent buffers.

Data-dependent acquisition (for spectral library building): The mass spectrometer was operated in data-dependent acquisition (DDA) mode, where the 30 most abundant precursor ions from the survey MS1 scan (250 ms) were isolated at 1 $m/z$ resolution for collision-induced dissociation tandem mass spectrometry (CID-MS/MS, 100 ms per MS/MS, 'high sensitivity' product ion scan mode) using the Analyst 1.7 (build 96) software with a total cycle time of 3.3 sec, as described[131].

Data-independent acquisitions: For quantification, all peptide samples were analyzed by data-independent acquisition (DIA, e.g. SWATH) using 64 variable-width isolation windows[132,133]. The window width was adjusted according to the complexity of the typical MS1 ion current observed within a certain $m/z$ range using a DIA 'variable window method' algorithm (more narrow windows were chosen in 'busy' $m/z$ ranges, wide windows in m/z ranges with few eluting precursor ions). DIA acquisitions produce complex MS/MS spectra, which are a composite of all the analytes within each selected Q1 $m/z$ window. The DIA cycle time of 3.2 s included a 250 ms precursor ion scan, followed by 45 ms accumulation time for each of the 64 variable SWATH segments.

## Mass-spectrometric data processing, quantification, and bioinformatics

MS data-dependent acquisitions (DDA) were analyzed using the data-base search engine ProteinPilot (SCIEX Beta 4.5, revision 1656) using the Paragon algorithm (4.5.0.0,1654) (specific results tables are uploaded to the data repository MassIVE). Using these results, a

MS/MS spectral library was generated in Spectronaut (Biognosys). The DIA/SWATH data was processed for relative quantification comparing peptide peak areas from various different time points (Supplementary Data 1). For DIA/SWATH MS2 data sets, quantification was based on XICs of 6-10 MS/MS fragment ions, typically y- and b-ions, matching to specific peptides present in the spectral libraries. Differential protein abundance analysis was performed using Student's *t*-test, and *p* values were corrected for multiple testing, specifically applying group-wise testing corrections using the Storey method[134]. Significantly changed proteins were accepted at a 5% FDR (*q* value <0.05), also see Supplementary Data 2.

### Western blotting

Conditioned media corresponding to the same number of cells were loaded for SDS−PAGE. Total cell and tissue lysates of mouse fibroblasts and intestine were prepared using RIPA lysis buffer (Thermo Fisher Scientific, #89900) with protease/phosphatase inhibitor (Thermo Fisher Scientific, #1861281). Protein concentration was measured with BCA protein assay kit (Thermo Fisher Scientific, #23227), and equal amount of proteins were loaded for SDS−PAGE. Proteins were transferred to PVDF membranes using iBlot 2 Dry Blotting System (Thermo Fisher Scientific). The membranes were blocked in 5% skim milk in Tris-buffered saline with 0.1% Tween20 (TBST) for 1 h and incubated in primary antibodies overnight at 4 °C followed by secondary antibodies for 1 h at room temperature. Blots were developed using the Pierce ECL Plus Western Blotting Substrate (Thermo Fisher Scientific, #32132). The uncropped and unprocessed scans of the blots were provided in a Source data file.

### TOPbrite dual-luciferase Wnt reporter assay

Generation of HEK293 cells stably expressing a TOPbrite firefly luciferase Wnt reporter and pRL-SV40 Renilla luciferase (HEK293-TB) was previously described[82]. HEK293-TB cells were maintained in DMEM/F12 (50:50), supplemented with 10% FBS, 2 mM Glutamax (Gibco, #35050-061) and 40 μg/ml hygromycin (Cellgro, #30-240-CR). For the luciferase reporter assay, HEK293-TB cells were seeded in 96-well plates (Falcon, #353377). The next day, cells were treated with indicated Wnt ligands (50 ng/ml) or CM diluted in culture media. After 24 h, the firefly and Renilla luciferase activity was measured using Dual-Glo Luciferase Assay system (Promega, #E2940) on a Perkin Elmer EnVision multilabel reader. The ratios of firefly luminescence to Renilla luminescence were calculated and normalized to control samples that were not treated.

### Phage display selection of peptide ligands against FZD5-CRD-Fc and FZD8-CRD-Fc

Phage pools of Linear-lib[82,135] were cycled through rounds of binding selections with FZD5-CRD-Fc or FZD8-CRD-Fc following established protocols[136]. We included 10 μM Herceptin in all rounds to block binders to the Fc-domain. After four rounds of binding selection, individual phage clones were analyzed in a high-throughput phage ELISA using plate-immobilized FZD-CRD-Fc as target or off-target for specificity as described previously[136]. The binding signal of the same phage particle to BSA was detected as non-specific binding. Clones with phage-binding signal to target over 0.5 and signal/noise ratio >10 were considered positive clones and subjected to DNA sequence analysis.

### Generation of inhibitor peptides and assessment of specificity and affinity of dFz5-43 and dFz8-80

Several inhibitor peptide candidates were synthesized and, for those that contained single cysteines, oxidized to their disulfide bridged dimers. Peptides were screened by surface plasmon resonance (SPR) against immobilized FZD7-Fc, FZD5-Fc and FZD8-Fc to assess their specificity and affinity. dFz5-43, the best hit from the FZD5 campaign showed binding to FZD5 in the 50 nM range, but at least 15-fold higher affinity (~800 nM) to FZD8 and no detectable binding to FZD7. The top

hits demonstrated exquisite selectivity to either FZD5/8 or FZD8, with affinities in the nM to pM range. dFz8-80, the best hit obtained in the FZD8 campaign showed \selective binding to only FZD8 and a dissociation constant of ~600 pM.

### Surface plasmon resonance

All measurements were done on a Biacore S200 instrument (GE Healthcare) with 10 mM Hepes pH 7.2, 150 mM NaCl and 0.001% P20 as running buffer. Fc-tagged FZD target proteins (R&D Systems) were immobilized on sensor spots of a Protein A Sensor chip (GE Health-care). A concentration series (3-fold dilution) across 6-dilutions of the indicated peptides were injected for single cycle kinetics. For Fab measurements, studies were performed with a CM5 chip and 50 mM Tris pH 8.0, 300 mM NaCl, 5% Glycerol and 0.05% Triton X as running buffer. Fabs were immobilized on chips using the Amine coupling kit (GE Healthcare, #BR100050) according to the manufacturer's instructions. A dilution series of the extracellular domain of recombinant mouse LRP5 (R&D Systems, #7344-LR) was injected. All data were double-referenced by subtracting the signal from a blank sensor spot and the signal from a buffer injection. Referenced data sets were fitted using the instrument's software. Peptides showing high specificity to either FZD5 and/or FZD8 were synthesized as disulfide bridged dimers (CSBio).

### ELISA

Proteins used in ELISAs were obtained from R&D Systems: PTK7-Fc (R&D Systems, #9799-TK), biotinylated Wnt3a (R&D Systems, #BT1324), biotinylated Wnt5a (R&D Systems, #BT645). His-tagged FZD7-CRD (FZD7-His) was produced in house as described[82]. Assay plates (384 Maxisorp plates, Nunc, # 464718) were prepared by incubation overnight with 30 μl/well Neutravidin (2 μg/ml, Thermo Fisher Scientific, #31055) in PBS at 4 °C. Plates were emptied and incubated with 100 μl/well of blocking buffer (PBS, 2% bovine serum albumin, 0.01% NaN₃) for 1 h at room temperature with shaking. Blocking buffer was aspirated and plates were washed 8x with 70 μl/well of PBS-T. In parallel, bio-Wnt3a or bio-Wnt5a were incubated in helper plates (384 well plates, Nunc; #264574) with a dilution series of PTK7-Fc or FZD7-His in the presence or absence of FZD7-His as a competitor in assay diluent (PBS + 0.5% BSA + 15 ppm ProClin 300 (Sigma-Aldrich, #48914-U) for 1 h at 4 °C. Protein mixtures (25 μl/well) were transferred from helper plates to 3 wells of assay plates (technical triplicates), and incubated for 1 h at 4 °C. Assay plates were aspirated and washed 8× with 70 μl/well of PBS-T. Binding was detected by 25 μl/well of anti-hIgG₁-Fc-HRP (0.2 μg/ml, Thermo Fisher Scientific, #A-10648) or anti-His-tag-HRP (0.2 μg/ml, Abcam, #ab1187) in assay diluent, incubated for 1 h at 4 °C. After washing, signal was developed by adding 25 μl/well of TMB substrate (KPL, #50-76-00) and incubation for up to 15 min. The reaction was stopped by adding 25 μl/well of 1 M phosphoric acid. Absorbance at 450 nm was recorded with a plate reader. Data were analyzed and fitted in GraphPad Prism 7.

### Live Ca2+ imaging of mouse intestinal organoids and data analysis

Organoids were live-imaged at identical-sized z stacks (2 micron intervals) using a 20X PlanFluor air objective attached to a 3i W Spinning Disc Confocal possessing a Zeiss AxioObserver M1 inverted microscope, Yokogawa W1 spinning disc unit and Photometrics Evolve EMCCD and Hamamatsu FLASH 4.0 sCMOS cameras. Z stacks were acquired every 10 s.

Z-stack images were converted to mean intensity projections and, where required; background was subtracted using a rolling ball algorithm. Segmentation was conducted on a max-projection over-time of the GCaMP6f channel using Cellpose's pre-trained "Cytoplasm" classifier[137]. Mean intensity traces were measured in each ROI over the time course in both the GCaMP6f and TdTomato channels using scikit-

image[138]. Ca²⁺ oscillation patterns were calculated in an unsupervised, automated manner using MATLAB. Mean intensity traces were smoothed using a Savitzky-Golay filter and peaks were identified using the "findpeaks" algorithm from the MATLAB Signal Processing Toolbox. Graphs for GCaMP6f/tdTomato fluorescence ratio, peak numbers, peak height, peak width and baselined area under the curve were generated using Python matplotlib and seaborn packages. Statistical analysis was performed using Python SciPy and scikit-post-hoc packages.

### RNAseq library preparation and sequencing

Total RNA was isolated using the RNeasy Plus Mini Kit (Qiagen, #74134) per manufacturer's instructions. Total RNA was quantified with Qubit RNA HS Assay Kit (Thermo Fisher Scientific, #Q32852) and quality was assessed using RNA ScreenTape on 4200 TapeStation (Agilent Technologies, #5067-5576). cDNA library was generated from 2 ng of total RNA using Smart-Seq V4 Ultra Low Input RNA Kit (Takara, #634894). 150 pg of cDNA was used to make sequencing libraries by Nextera XT DNA Sample Preparation Kit (Illumina, #FC-131-1024). Libraries were quantified with Qubit dsDNA HS Assay Kit (Thermo Fisher Scientific, #Q32851) and the average library size was determined using D1000 ScreenTape on 4200 TapeStation (Agilent Technologies, #5067-5582). Libraries were pooled and sequenced on NovaSeq 6000 (Illumina) to generate 30 millions single-end 50-base pair reads for each sample.

### RNAseq analysis

Sequencing reads were filtered and aligned using HTSeqGenie v4.2.2. GSNAP v2013-11-01 was used for alignment, through the HTSeqGenie wrapper, against the mouse reference genome (NCBI Build 38). Reads within coding sequences were counted to determine an estimate of gene expression. Read counts were scaled by library size, quantile normalized and precision weights were calculated using the voom R package[139]. Differential expression analysis was performed using the limma package[140]. Differential gene expression was calculated using voom+limma with multiple-hypothesis correction of $P$ values performed using the Benjamini-Hochberg method. Cutoff thresholds of fold change >1.5 and adjusted $p$ value <0.05 were used. Transcription factor binding motif analysis was performed using RcisTarget[141].

### single-nucleus RNAseq (snRNA-seq)

Organoids were harvested, washed in PBS, pelleted, and frozen on dry ice for snRNA-seq. Nuclei were isolated following the protocol described previously[142]. Briefly, frozen samples were resuspended in TST buffer (146 mM NaCl, 10 mM Tris, 1 mM CaCl₂, 21 mM MgCl₂, 0.03% Tween20, and BSA) and pelleted by centrifugation. The nuclei pellets were resuspended in ST buffer (146 mM NaCl, 10 mM Tris, 1 mM CaCl₂, 21 mM MgCl₂), passed through a 35 μm filter, and counted using a hemocytometer. Generation of gel Beads-in-emulsion, cDNA amplification, and library preparation were performed with Chromium Next GEM Single Cell 3′ Reagent Kits v3.1 (10x Genomics) following the manufacturer's instructions. 8000-10,000 nuclei were loaded per channel of a 10x Genomics chip.

### single-nucleus RNAseq analysis

Reads were aligned to the mouse genome (mm10/Ensembl 98) using the 10x CellRanger 'count' function with the resulting matrix imported into the scanpy framework[143]. For each sample, we discarded cells falling more than 3 MAD (mean absolute deviation) above/below the population in any of three metrics: Fraction of mitochondrial reads (above), total number of reads (below), or total number of features detected (below). Library sizes per cell were normalized to a target sum of 1e4 using the 'normalize_total' function from scanpy followed by log-transformation. For projection and cluster identification, data were restricted to highly variable genes using the default parameters in scanpy's 'pp.highly_variable_genes' module and the data scaled to unit variance and zero mean after correcting data for differences in mitochondrial read counts and total counts using linear regression and followed by dimensionality reduction using PCA. Data were then integrated at the level of individual sample/library using Harmony Clusters were identified using the Leiden algorithm implemented in scanpy following calculation of the neighborhood graph of the observed cell data using the top 25 PCs and a local neighborhood of 10 for manifold approximation.UMAP and tSNE projections were subsequently calculated using default parameters[144].

The above processing resulted in 14 clusters with no evidence of sample-level batch effects. These clusters were subsequently annotated by examining the expression of markers from major intestinal cell types (see Figure S1e). To further support this annotation, we calculated marker genes using the scoreMarkers function from the scran package in R as applied to the log-transformed, normalized count data[145]. To calculate gene scores for gene sets with many genes (e.g. in the case of the YAP target genes[99]) we made use of the AddModuleScore function from Seurat[146]. The list of genes used for ISC/CBC signature - Lgr5, Olfm4, Ascl2, Axin2, Hmgb2, Smoc2, Lrig1. This function calculates the average expression level of the genes in a gene set normalized by the expression of an expression-level matched set of control genes[147]. All subsequent statistical analyses were carried out in R as described in the text.

### Statistics and reproducibility

Data were presented as means ± SD, and the n value means biological replicates with individual values representing a mouse, unless otherwise indicated. Statistical analyses were performed using GraphPad Prism 9.0 (GraphPad Software, Inc., La Jolla, CA, USA). For comparisons between two independent groups, two-tailed unpaired Student's t test was used. Statistical significance of differences between multiple groups was determined using one-way ANOVA with Tukey's post-hoc multiple comparison tests. Experiments were independently repeated with similar results. For Figs. 1d, q, 2g, i, 3a, d, i and Supplementary Figs. 2a, d–e, 3b, and 7c the experiments were repeated twice. For Figs. 1a, 2f, 5c, 7b–f and Supplementary Fig. 5e, the experiments were repeated 3 times. For Fig. 7a and Supplementary Fig. 7a, the experiment was repeated 4 times. For Fig. 5b, the experiment was repeated five times.

### Materials

See Supplementary Information for tables listing TaqMan assay probes for qPCR (Supplementary Table 1), antibodies (Supplementary Table 2), recombinant proteins (Supplementary Table 3), and compounds and inhibitors (Supplementary Table 4) used in this study.

### Reporting summary

Further information on research design is available in the Nature Portfolio Reporting Summary linked to this article.

## Data availability

The bulkRNA-seq and snRNA-seq raw data generated in this study have been deposited in the GEO database under accession code GSE219128. The mass-spectrometric raw data are deposited at ProteomeXchange with the ID PXD028176. Other data generated in this study are provided in the Supplementary Information and Source Data file. Source data are provided with this paper.

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

## Acknowledgements

We acknowledge the support of instrumentation from the NCRR shared instrumentation grant 1S10 OD016281 (Buck Institute). This work was supported by a grant from the National Institute on Aging (U01 AG060906-02, PI: Schilling).

## Author contributions

J.Y. conceived of the project, performed most experiments with organoids and senescent cells, analyzed the data, coordinated collaborations and wrote the manuscript, Y.Z., L.Z., S.H., and A.H.N. developed, characterized and tested Fzd binding peptides and performed in vitro binding assays between Ptk7 and Wnt ligands/receptors, O.M. and A.J.K. performed and analyzed Ca2+ live imaging in organoids, O.M. performed TF enrichment analysis on RNAseq data, D.T.M. and C.P.L. performed characterization of SCM-induced organoid phenotype and biochemical fractionation of SCM, C.W. and B.S. performed MassSpec analysis and identification of SASP factors, Y.L. and Z.M. performed RNA sequencing, M.B.C. contributed protocols for snRNA-seq of intestinal organoids, CCO and DG performed scRNAseq analysis, J.C. provided insight into senescence and the SASP, R.N.H. and H.J. directed the project, planned and interpreted experiments and wrote the manuscript.

## Competing interests

J.Y., M.B.C., R.N.H., Y.Z., L.Z., C.C.O., D.G., H.J. are employees and/or stock holders of Genentech Inc. The remaining authors declare no competing interests.
