## [Peer Review File · Nature Communications]

Senescent cells perturb intestinal stem cell differentiation through Ptk7 induced noncanonical Wnt and YAP signalingREVIEWER COMMENTS

Reviewer #1 (Remarks to the Author):

In this study the authors use intestinal organoids as a model to study the impact of senescent cells on tissue regeneration. They use senescent MEFs as a source of the senescence-associated secretory phenotype (SASP) and show that stem cell differentiation in these organoids is impaired by the secretome of senescent cells. The impact is direct as immune cells are not present in this in vitro system. Using mass spectrometry, sPKT7 was identified as one of the abundant SASP components in conditioned media from senescent cells. The authors demonstrate that sPKT7 is involved in disrupting organoid development by virtue of transcriptional activation of YAP/TEAD. Furthermore, PKT7 is shown to signal through non-canonical WNT pathway (WNT5a-FZD7). Ca⁺⁺ level oscillations are increased upon treatment of intestinal stem cells with WNT5a and, in the presence of senescent cell conditioned media, drives aberrant organoid morphology.

Overall, this is a well written and clear manuscript that provides compelling evidence of the role of sPKT7 as a SASP factor in affecting stem cell differentiation. Strengths of the manuscript include i) using two inducers of senescence (IR and doxorubicin); ii) rigorous demonstration of the signaling mechanism elicited by sPKT7 using genetics, multiple pharmacologic inhibitors, direct administration of recombinant protein, and antibody depletion. The impact of the body of work is that it identifies a mechanism by which SASP has adverse effects on tissue regeneration, which is contrary to the generally accepted thought that SASP is necessary for wound healing. In addition, a potent biologic function is ascribed to a single SASP factor.

That being said, the experiments performed were all done in vitro, which while essential for teasing out molecular mechanism, leave questions about relevance in vivo.

To nail down the upstream mechanism, it would be valuable to detect PKT7 on the surface of senescent cells vs. non-senescent cells to determine if senescence leads to upregulation of this kinase or simply the proteases that release sPKT7 from the cell surface.

Is increased sPKT7 detected in the serum of old mice which have a high abundance of senescent cells? Is PKT7, MMP14 or ADAM17 expression increased in large intestine from old mice? The conditioned media from senescent cells changed the ratio of differentiated intestinal cells in organoids. Is this shift in cell ratios known to occur with aging in mice?

To further establish physiological relevance, is it possible to seed senescent cells into the organoid cultures and determine the impact on organoid morphology?

While the others did a thorough job in illustrating that the effect of conditioned media (CM) from senescent cells (SnC) differs from that of quiescent cells (QC) by swapping the media and observing a consistent correlation of cystic morphology associated with SnC-CM, one messy variable remains: serum. The cells are driven to quiescence by serum deprivation. It would be valuable to test the impact of QC-CM spiked with serum and SnC-CM with serum deprivation on the morphology of intestinal organoids.

Additionally, the authors demonstrate a change in organoid morphology when the CM are switched but does this correspond with a change in the signaling events? Measuring this will further establish cause and effect.

Since sPKT7 is not secreted by senescent cells but cleaved from the surface of the cells via a SASP factor, technically sPKT7 is not a SASP factor. Adhering to this strict definition in fact increases the impact of this body of work, because in fact the phenotype of senescent cells includes a secretome as well as surface remodeling to release more bioactive peptides.

The SASP is known to drive secondary senescence. Is there any evidence of senescence in the organoid cultures treated with SnC-CM?

Minor:

Figure 7b is confusing because two mechanisms are illustrated on what looks like a single cell. Consider illustrating the signaling events of a healthy intestinal stem cell and another exposed to sPKT7.

Words inserted or missing words on line 64 and 108.

Reviewer #2 (Remarks to the Author):

In this manuscript, Jasper's lab found that conditioned medium from senescent MEFs impairs cell differentiation in mouse intestinal organoids via secreted Ptk7. Secreted Ptk7 acts through the non-canonical Wnt signaling pathway and cytosolic Ca²⁺ oscillation to induce the phenotype. They did transcriptome analysis and found that YAP/TEAD target genes are enriched in SASP-exposed organoids and inhibition of YAP nuclear translocation rescued cystic organoid morphology. They concluded that Ptk7 secreted by senescent cells is a critical mediator of intestinal stem cell dysfunction in aging. The strength of the paper is the identification of the Ptk7/FZD/Ca²⁺/YAP signaling as a mechanism for senescent cells to regulate ISC aging. However, the lack of biological relevance dampens the enthusiasm to publish it in Nat Commun.

Major concerns:

1. I do not understand why the authors chose MEFs for the study. The most suitable would be primary intestinal stromal cells, which can be easily isolated and cultured. Also, I do not understand why the authors used X-ray or doxo to induce cell senescence. Replicative senescence is definitely more relevant. I suggest that the authors use replicative senescent intestinal stromal cells to verify their key findings.
2. It is not clear what is "cystic organoid phenotype". Do the authors see this in mice radiated, mice treated with Doxo, or naturally aged mice?
3. The work lacks biological relevance. Do they see expression of Ptk7 in the intestines of aged mice? Do they see activation of Ca²⁺/YAP in the ISCs of aged mice?
4. The authors fail to discuss cell-autonomous mechanisms controlling ISC aging.

Minor points:

1. in Figure 1, since the sizes of the organoids differ greatly in normal and cystic organoids, it is better to normalize proliferating cells to total number of cells.
2. Figure 2J. Loading control is needed.
3. Figure 5. In RNAseq analysis, did the authors spot any genes that regulate ISC differentiation?

Reviewer #3 (Remarks to the Author):

In this study, Yun et al. examine the conditioned media (CM) of mouse embryonic fibroblasts (MEFs) exposed to x-ray irradiation (defined as 'senescent') vs. MEFs that underwent serum withdrawal (defined as 'quiescent') for effects on intestinal organoid morphology. They find that 'senescent' CM induces a cystic morphology relative to 'quiescent' CM and identify the soluble, N-terminal fragment of the Ptk7 receptor tyrosine kinase, which can be cleaved from the cell surface by extracellular enzymes including MMPs, as sufficient to induce the cystic organoid morphology. The authors provide evidence that N-terminal Ptk7 mediates its effects via the potentiation of Fzd-dependent, non-canonical Wnt signaling. Ptk7 was observed to bind Fzd7 and Wnt3a/Wnt5a. Further, Wnt5a/Ptk7-dependent Ca²⁺ oscillations (known to occur downstream of non-canonical Wnt pathway activity) were described, and inhibition of these oscillations inhibited Wnt5a/Ptk7-driven changes in organoid morphology (from budding to cystic). Transcriptome profiling ultimately demonstrated Yap activation downstream of Wnt5a/Ptk7, and similarly Yap activity was found to be important for cystic organoid formation.

Overall this study is technically sound and it uncovers a novel molecular pathway linking non-canonical Wnt pathway activation with Yap activity. My questions/concerns primarily surround the underlying conceptual premise.

The first major concern I have with this study is that the singular functional readout throughout the experimentation is a change in organoid morphology from budding to cystic. It is not clear to me (or the field at large as far as I know) what 'cystic' morphology really means. A lot of things induce cystic

organoid morphology, e.g., the cell of origin in organoid cultures (e.g., PMC6008251, PMID24139799), Apc inactivation, or any number of things, many of which elicit different responses in vivo. Thus, there is no consensus on what 'cystic' organoid morphology means for the biology of the intestinal epithelium. For example, Yap activation in this study is associated with decreases in secretory lineage gene expression with no changes to the canonical crypt base columnar marker Lgr5. A number of recent studies have identified Yap as a critical mediator of a fetal intestinal stem cell/post-injury regenerative transcriptional program, and fetal intestinal stem cells with this Yap program activated grow as cysts (PMC6042247, PMC5766831)

In vivo, this regenerative fetal program is associated with a complete silencing of Lgr5 and the canonical crypt base columnar stem cell gene expression program (e.g., PMC5766831). The current study implicates N-terminal Ptk7 as a senescence-associated molecule capable of Yap activation. There is a preponderance of evidence that Yap activation in the epithelium is associated with proliferative regeneration. In contrast, during ageing, epithelial proliferation and regeneration are compromised. Given that the expectation is an increase in senescent cells as a function of time, the conceptual model presented in this study do not appear to align with established in vivo biology. Thus, while the molecular biology presented in this study is novel and technically sound, the context in which it has been described has no clear bearing on normal intestinal epithelial biology. Some questions that arise related to this:

Is the Lgr5+/CBC gene signature maintained in Ptk7-driven cystic organoids?
Is the fetal ISC signature enriched? This would be expected based on the Yap activation
And, most importantly, what indication is there that Ptk7 plays any role in the intestine?

Ptk7 was identified from irradiated MEFs. While the study elegantly elucidates the effects of N-terminal Ptk7 on organoid morphology, there is no reason to believe that irradiate MEFs have anything to do with intestinal biology or a SASP program in that tissue. Is Ptk7 expressed in the epithelium or the subepithelial telocytes that provide Wnt ligands to the stem cells? Does Ptk7 N-terminal cleavage increase as a function of age in the intestine? Or in senescent cells in vivo? I think these critical points need to be addressed for a journal of this caliber.

Minor comments:

-what is the expression of Ptk7 in MEFs vs. organoids? Can the authors show the ratio of total to cleaved N-terminal Ptk7, perhaps by Western?

-what is the effect of Ptk7 treatment on long-term stem cell activity- e.g., organoids cultured in the presence of N-terminal Ptk7 digested down to single cells and replated serially over several passages would give an indication of the degree of stem cell self-renewal happening in the cultures.

-A number of studies have described the effects of Wnt5a/non-canonical Wnt signaling on the development of the intestinal epithelium. The findings in the current study should be discussed in relation to these prior studies.

Response to Reviewers – Yun et al., NCOMMS-21-45153-A

We would like to thank the three reviewers for their positive reception of our work. Reviewer 1 points out that our study ‘provides compelling evidence of the role of sPKT7 as a SASP factor in affecting stem cell differentiation’, Reviewer 2 highlights as a particular strength ‘the identification of the Ptk7/FZD/Ca²⁺/YAP signaling as a mechanism for senescent cells to regulate ISC aging’, and Reviewer 3 highlights that ‘overall this study is technically sound and it uncovers a novel molecular pathway linking non-canonical Wnt pathway activation with Yap activity.’

But the reviewers also highlight a few concerns, ranging from a lack of *in vivo* data to specific experimental questions. We have now performed a range of additional studies to allay these concerns and believe that our study has significantly improved and should now satisfy the reviewers’ reservations.

Our new data show that:

- Ptk7 is expressed in intestinal fibroblasts
- the secretion of the N-terminal domain of Ptk7 increases in the aging mouse intestine.
- Single-nuclear RNASeq of organoids exposed to senescent conditioned media confirms changes in cell composition and identifies a cell population expressing a fetal gene signature, further supporting the role of Yap in promoting the cystic organoid phenotype.
- Co-culturing of organoids with senescent intestinal fibroblasts further supports the role of fibroblast-derived sPtk7 in the induction of the cystic phenotype

We have marked the edited sections in the manuscript text using a blue font. In the following, we provide a point – by – point response to the issues raised by the reviewers:

Reviewer #1 (Remarks to the Author):

In this study the authors use intestinal organoids as a model to study the impact of senescent cells on tissue regeneration. They use senescent MEFs as a source of the senescence-associated secretory phenotype (SASP) and show that stem cell differentiation in these organoids is impaired by the secretome of senescent cells. The impact is direct as immune cells are not present in this in vitro system. Using mass spectrometry, sPTK7 was identified as one of the abundant SASP components in conditioned media from senescent cells. The authors demonstrate that sPTK7 is involved in disrupting organoid development by virtue of transcriptional

activation of YAP/TEAD. Furthermore, PTK7 is shown to signal through non-canonical WNT pathway (WNT5a-FZD7). Ca⁺⁺ level oscillations are increased upon treatment of intestinal stem cells with WNT5a and, in the presence of senescent cell conditioned media, drives aberrant organoid morphology.

Overall, this is a well written and clear manuscript that provides compelling evidence of the role of sPKT7 as a SASP factor in affecting stem cell differentiation. Strengths of the manuscript include i) using two inducers of senescence (IR and doxorubicin); ii) rigorous demonstration of the signaling mechanism elicited by sPKT7 using genetics, multiple pharmacologic inhibitors, direct administration of recombinant protein, and antibody depletion. The impact of the body of work is that it identifies a mechanism by which SASP has adverse effects on tissue regeneration, which is contrary to the generally accepted thought that SASP is necessary for wound healing. In addition, a potent biologic function is ascribed to a single SASP factor.

We'd like to thank the reviewer for the positive reception of our work.

That being said, the experiments performed were all done in vitro, which while essential for teasing out molecular mechanism, leave questions about relevance in vivo.

To nail down the upstream mechanism, it would be valuable to detect PKT7 on the surface of senescent cells vs. non-senescent cells to determine if senescence leads to upregulation of this kinase or simply the proteases that release sPKT7 from the cell surface.

We thank the reviewer for the suggestion. We are now including new data showing that cellular senescence results in strongly increased MMP14 levels in fibroblasts and a corresponding increase in shedding of Ptk7, while Ptk7 transcript levels (measured by qPCR) are not affected significantly. (Figure 3i and Supplementary Figure 2h).

Is increased sPKT7 detected in the serum of old mice which have a high abundance of senescent cells? Is PKT7, MMP14 or ADAM17 expression increased in large intestine from old mice? The conditioned media from senescent cells changed the ratio of differentiated intestinal cells in organoids. Is this shift in cell ratios known to occur with aging in mice?

We thank the reviewer for these questions. While detection of Ptk7 in the mouse serum was unsuccessful (at any age), we did detect increased sPtk7 in intestinal extracts from old mice, supporting the idea that sPtk7 is a component of the SASP that is increased in an age-related manner. These data also indicate that sPtk7 may act in a local fashion in the tissue rather than being released into the serum. These additional data are described in Figure 2i, j and discussed in the new discussion.

We also investigated the expression of the more than 70 ADAM and MMP family members in the aging mouse intestine, but did not observe a significant change in ADAM and MMP gene expression in whole-intestinal extracts of aging mice. Based on our related study in flies (Hu et al., 2021), we anticipate that MMPs are induced in a specific set of senescent cells, and that extensive single-cell analysis of old intestines will have to be performed to assess whether MMP induction in specific cells contributes to the increase in sPTK7 in aging intestines.

Changes in cell type ratios in the aging mouse intestine have not been fully characterized, with often contradicting reports in the literature (Moorefield, Andres et al. 2017, Nalapareddy, Nattamai et al. 2017, Mihaylova, Cheng et al. 2018, Igarashi, Miura et al. 2019, Pentinmikko, Iqbal et al. 2019, Sovran, Hugenholtz et al. 2019, Gebert, Cheng et al. 2020, Sirvinskas, Omrani et al. 2022). As such, age-related changes result from a combination of intrinsic changes in stem cell function and extrinsic factors impacting stem cell activity, differentiation capacity, and daughter cell differentiation. Further extensive studies are needed to develop a consensus on the impact of aging on the intestinal epithelium.

To further establish physiological relevance, is it possible to seed senescent cells into the organoid cultures and determine the impact on organoid morphology?

We thank the reviewer for this suggestion. We are now including new data showing that co-culture of intestinal organoids with senescent primary intestinal fibroblasts recapitulates the cystic organoid morphology caused by senescent conditioned media (Figure 3j, k). Importantly, this phenotype is also dependent on Ptk7, as blocking Ptk7 using an antibody also inhibits cystic organoid formation.

While the others did a thorough job in illustrating that the effect of conditioned media (CM) from senescent cells (SnC) differs from that of quiescent cells (QC) by swapping the media and observing a consistent correlation of cystic morphology associated with SnC-CM, one messy

variable remains: serum. The cells are driven to quiescence by serum deprivation. It would be valuable to test the impact of QC-CM spiked with serum and SnC-CM with serum deprivation on the morphology of intestinal organoids.

We believe there is a misunderstanding here and would like to clarify how we prepared conditioned media. We washed quiescent and senescent cells with serum-free DMEM twice and added serum-free DMEM 24 hours before collection of conditioned media. Both quiescent and senescent conditioned media do not contain any serum, which eliminates the possibility of serum affecting the morphology of intestinal organoids. We have updated the description of this in the materials and methods for more clarity.

Additionally, the authors demonstrate a change in organoid morphology when the CM are switched but does this correspond with a change in the signaling events? Measuring this will further establish cause and effect.

We thank the reviewer for the suggestion. We now include data showing activation or inhibition of YAP in the corresponding conditions (Supplementary Figure 7c).

Since sPKT7 is not secreted by senescent cells but cleaved from the surface of the cells via a SASP factor, technically sPKT7 is not a SASP factor. Adhering to this strict definition in fact increases the impact of this body of work, because in fact the phenotype of senescent cells includes a secretome as well as surface remodeling to release more bioactive peptides.

We have now included the reviewer's point in our discussion. For simplicity, we are still referring to sPtk7 as a component of the SASP, as we the SASP can be defined as the collection of bioactive factors released from senescent cells and detected in their supernatant. But we can of course change this in the text if the reviewer believes that this definition is incorrect.

The SASP is known to drive secondary senescence. Is there any evidence of senescence in the organoid cultures treated with SnC-CM?

We have tested this idea by assessing if senescence could be detected in organoids treated with SCM. We find that SASP factors from senescent conditioned media does not drive secondary senescence, as immunostaining for pH3 and Ki67 shows no cell cycle arrest in organoids exposed to SCM (Supplementary Figure 1b). This is consistent with the data included in our original submission, which indicate that the SCM – induced phenotype is reversible, i.e. that organoids start budding after SCM is removed, further suggesting that SCM treated cells do not enter an irreversible senescent state. We now also show that SCM has no long-term effect on stem cell proliferative capacity: When organoids exposed to SCM were dissociated and passaged, and maintained in regular organoid culture media, colony forming efficiency remained comparable to those exposed to QCM (Supplementary Figure 1f, g).

Minor:

Figure 7b is confusing because two mechanisms are illustrated on what looks like a single cell. Consider illustrating the signaling events of a healthy intestinal stem cell and another exposed to sPKT7.

Words inserted or missing words on line 64 and 108.

We thank the reviewer for the suggestions. We have made changes as suggested.

Reviewer #2 (Remarks to the Author):

In this manuscript, Jasper's lab found that conditioned medium from senescent MEFs impairs cell differentiation in mouse intestinal organoids via secreted Ptk7. Secreted Ptk7 acts through the non-canonical Wnt signaling pathway and cytosolic Ca²⁺ oscillation to induce the phenotype. They did transcriptome analysis and found that YAP/TEAD target genes are enriched in SASP-exposed organoids and inhibition of YAP nuclear translocation rescued cystic organoid morphology. They concluded that Ptk7 secreted by senescent cells is a critical mediator of intestinal stem cell dysfunction in aging. The strength of the paper is the identification of the Ptk7/FZD/Ca²⁺/YAP signaling as a mechanism for senescent cells o regulate ISC aging. However, the lack of biological relevance dampens the enthusiasm to publish it in Nat Commun.

We'd like to thank the reviewer for the positive comments on our work and for the critical assessment of impact. We have now included additional data that indicate a relevance of the described mechanism on intestinal aging in vivo.

Major concerns:

1. I do not understand why the authors chose MEFs for the study. The most suitable would be primary intestinal stromal cells, which can be easily isolated and cultured. Also, I do not understand why the authors used X-ray or doxo to induce cell senescence. Replicative senescence is definitely more relevant. I suggest that the authors use replicative senescent intestinal stromal cells to verify their key findings.

We thank the reviewer for the suggestion. We have now included new data using primary intestinal fibroblasts as suggested, showing that senescent conditioned media from these fibroblasts results in the same sPtk7-dependent effects as SCM from MEFs (Figure 3i-k and Supplementary Figure 2g). We agree that replicative senescence is another paradigm for senescence, but we don't agree that X-ray or Doxorubicin-induced senescence are not physiologically relevant. The induction of senescence by irradiation or chemotherapy, as well as more generally by DNA damage, has important consequences for patients with chemotherapy or radiation therapy and understanding the consequences of the increase in senescent cells after such treatments is critical. Practically, replicative senescence of intestinal fibroblasts is not as homogeneous and robust as X-ray or Doxo induced senescence. We therefore chose these paradigms to test the effects of SASP on intestinal stem cells.

2. It is not clear what is "cystic organoid phenotype". Do the authors see this in mice radiated, mice treated with Doxo, or naturally aged mice?

We apologize that this was not clearly explained and characterized. We are now including a single nuc-Seq analysis of the cystic organoids and further characterization of the phenotype. Our data confirm that the cystic organoids are a consequence of a lack of stem cell differentiation, with an increase in a cell population expressing a 'fetal' signature, which is a gene signature regulated by

YAP. This is consistent with our interpretation that the cystic phenotype is caused by activation of non-canonical Wnt signaling resulting in increased nuclear localization of YAP.

Interestingly, activation of YAP has also been shown to be a consequence of excessive tissue damage in the intestinal epithelium, where YAP causes reversion of epithelial cells to a de-differentiated state that is characterized by the expression of a fetal signature and is required for proper tissue repair (Yui, Azzolin et al. 2018). Furthermore, non-canonical Wnt signaling has been shown to be required for tissue repair after extensive epithelial damage in the intestine, suggesting that the mechanism we described in this study may play a role in such tissue repair paradigms. Future work will examine that role.

We now also include data showing that Ptk7 is expressed in fibroblasts in the mouse intestine, and that the secreted N-terminal domain of Ptk7 is detected at higher levels in the intestine of old mice, supporting the idea that aging is associated with increased SASP levels, including increased shedding of sPtk7 from intestinal fibroblasts.

Of note, we have now also published a related study in flies, in which the *Drosophila* Ptk7 orthologue Otk was found to be required for intestinal stem cell migration to the site of injury (Hu et al, Nature Commun. 2021). In flies, the N-terminal domain of Otk is shed from enteroendocrine cells by Mmp action in response to epithelial injury and activates non-canonical Wnt signaling in stem cells. The parallels to our study are striking and suggest an evolutionarily conserved mechanism for tissue regeneration. We are evaluating this mechanism during regeneration of the mouse intestine now, but these experiments are beyond the scope of the work presented here.

3. The work lacks biological relevance. Do they see expression of Ptk7 in the intestines of aged mice? Do they see activation of Ca²⁺/YAP in the ISCs of aged mice?

We thank the reviewer for pointing this out. We have now included new data showing that Ptk7 is expressed in intestinal fibroblasts and that aging is associated with increased shedding of Ptk7 in the intestine, as the processed N-terminal domain of Ptk7 is detected at higher levels in the old intestine (Figure 2i, j). Assessing changes in Ca²⁺ signaling in the crypts of old mice would be very interesting, but requires the aging for 24 months of a cohort of mice carrying the Ca²⁺ reporter. We hope that the reviewer agrees that this would be beyond the scope of the current study, but certainly an interesting study to follow up with. We did assess YAP nuclear localization in the intestinal epithelium of old mice, but detection of YAP by immunostaining in these tissues was not conclusive (weak and heterogeneous signal).

4. *The authors fail to discuss cell-autonomous mechanisms controlling ISC aging.*

We apologize for this omission and are now starting our discussion section with a new paragraph summarizing the reported cell-autonomous and non-autonomous aging phenotypes of intestinal stem cells.

Minor points:

1. *in Figure 1, since the sizes of the organoids differ greatly in normal and cystic organoids, it is better to normalize proliferating cells to total number of cells.*

We thank the reviewer for pointing this out. We have now added the number of proliferating cells normalized to total number of cells (Supplementary Figure 1c).

2. *Figure 2J. Loading control is needed.*

For conditioned media, loading controls such as Actin or Tub cannot be used as only secreted proteins exist in the media. Thus, we loaded conditioned media corresponding to the same number of cells. In addition, we included new results from primary intestinal fibroblasts that show increased sPtk7 in conditioned media and decreased full-length Ptk7 in the cell pellet (detecting Tubulin as a loading control), further strengthening the notion that the N-terminal domain of Ptk7 is shed from senescent fibroblasts (Figure 3i). Our data from old intestines (Fig. 2i) further supports increased shedding of sPtk7 (using Actin as loading control).

3. *Figure 5. In RNAseq analysis, did the authors spot any genes that regulate ISC differentiation?*

Our experimental design was to detect early transcriptomic changes, thus we collected samples two days after exposure to conditioned media. We detected overall very low expression of genes that regulate ISC differentiation at the time point of collection, and most of them did not pass the cutoff p-value for significance in differential gene expression. We do, however, observe reduced expression of *Neurog3* and *Gfi1* when exposed to SCM. We included these data in

(Supplementary Figure 6h, i). Consistently, our new snRNA-seq data from organoids exposed to QCM and SCM show reduced enteroendocrine progenitors and differentiated secretory cell types after SCM exposure (Figure 1n-p and Supplementary Figure 1e, f).

Reviewer #3 (Remarks to the Author):

In this study, Yun et al. examine the conditioned media (CM) of mouse embryonic fibroblasts (MEFs) exposed to x-ray irradiation (defined as 'senescent') vs. MEFs that underwent serum withdrawal (defined as 'quiescent') for effects on intestinal organoid morphology. They find that 'senescent' CM induces a cystic morphology relative to 'quiescent' CM and identify the soluble, N-terminal fragment of the Ptk7 receptor tyrosine kinase, which can be cleaved from the cell surface by extracellular enzymes including MMPs, as sufficient to induce the cystic organoid morphology. The authors provide evidence that N-terminal Ptk7 mediates its effects via the potentiation of Fzd-dependent, non-canonical Wnt signaling. Ptk7 was observed to bind Fzd7 and Wnt3a/Wnt5a. Further, Wnt5a/Ptk7-dependent Ca⁺ oscillations (known to occur downstream of non-canonical Wnt pathway activity) were described, and inhibition of these oscillations inhibited Wnt5a/Ptk7-driven changes in organoid morphology (from budding to cystic). Transcriptome profiling ultimately demonstrated Yap activation downstream of Wnt5a/Ptk7, and similarly Yap activity was found to be important for cystic organoid formation.

Overall this study is technically sound and it uncovers a novel molecular pathway linking non-canonical Wnt pathway activation with Yap activity. My questions/concerns primarily surround the underlying conceptual premise.

We'd like to thank the reviewer for the positive reception of our work and have now addressed the remaining conceptual concerns as discussed below in the point-by-point response.

The first major concern I have with this study is that the singular functional readout throughout the experimentation is a change in organoid morphology from budding to cystic. It is not clear to me (or the field at large as far as I know) what 'cystic' morphology really means. A lot of things induce cystic organoid morphology, e.g., the cell of origin in organoid cultures (e.g., PMC6008251,

PMID24139799), Apc inactivation, or any number of things, many of which elicit different responses in vivo. Thus, there is no consensus on what 'cystic' organoid morphology means for the biology of the intestinal epithelium. For example, Yap activation in this study is associated with decreases in secretory lineage gene expression with no changes to the canonical crypt base columnar marker Lgr5. A number of recent studies have identified Yap as a critical mediator of a fetal intestinal stem cell/post-injury regenerative transcriptional program, and fetal intestinal stem cells with this Yap program activated grow as cysts (PMC6042247, PMC5766831)

We agree, and in fact have now observed an increase in cells expressing that fetal signature when analyzing SCM-exposed organoids using snRNAseq (Supplementary Figure 6f, g). Our data is thus consistent with the studies cited by the reviewer, demonstrating that sPtk7, shed by senescent cells, causes a cellular response in intestinal epithelial cells that is associated with an injury response. This is also consistent with the observation by other groups that senescent cells are associated with repair responses during acute injury, and that their accumulation in aging tissues induces aberrant physiological conditions by activating that injury response chronically (Collado, Blasco et al. 2007, Jun and Lau 2010, Demaria, Ohtani et al. 2014, Munoz-Espin and Serrano 2014).

Of note, we have now also published a related study in flies, in which the *Drosophila* Ptk7 orthologue Otk was found to be required for intestinal stem cell migration to the site of injury during regeneration (Hu et al, Nature Commun. 2021). In flies, the N-terminal domain of Otk is shed from enteroendocrine cells by Mmp action in response to epithelial injury and activates non-canonical Wnt signaling in stem cells. The parallels to our study are striking and suggest an evolutionarily conserved mechanism for tissue regeneration. Sustained activation of this mechanism in the aging intestine due to the accumulation of senescent cells may contribute to tissue dysfunction. We show now that elevated levels of sPtk7 can indeed be detected in the aging intestine (Fig. 2i,j).

In vivo, this regenerative fetal program is associated with a complete silencing of Lgr5 and the canonical crypt base columnar stem cell gene expression program (e.g., PMC5766831).

The current study implicates N-terminal Ptk7 as a senescence-associated molecule capable of Yap activation. There is a preponderance of evidence that Yap activation in the epithelium is associated with proliferative regeneration. In contrast, during ageing, epithelial proliferation and regeneration are compromised. Given that the expectation is an increase in senescent cells as a

function of time, the conceptual model presented in this study do not appear to align with established in vivo biology. Thus, while the molecular biology presented in this study is novel and technically sound, the context in which it has been described has no clear bearing on normal intestinal epithelial biology.

We understand the reviewer's point and have now edited the discussion and introduction to better explain our reasoning. Of note, we fully agree with the reviewer that Yap activation promotes intestinal repair after acute injury. The fact that aging is associated with compromised epithelial proliferation is less clearly established, however. Based on several reports, aging is associated with a decline in Wnt signaling in the crypt niche, resulting in a reduction in Lgr5+ cell numbers (Nalapareddy, Nattamai et al. 2017, Pentimikko, Iqbal et al. 2019). EdU incorporation in the crypt is not significantly reduced in old mice, however (Moorefield, Andres et al. 2017). In addition, aging is associated with a significant increase in colorectal cancer incidence, a phenotype that may be caused by increased inflammation (Roulis, Kaklamanos et al. 2020). Our organoid data suggest that YAP activation induced by sPtk7 leads to defects in cell differentiation, especially secretory cell types, which has been observed in the mouse intestine (Zhou, Zhang et al. 2011, Imajo, Ebisuya et al. 2015). It can be anticipated that chronic activation of YAP in the intestinal epithelium by the elevated levels of sPtk7 which we now report are found in the aging intestine (Figure 2i, j), result in age-related differentiation defects and potentially even in increased cancer incidence. Subsequent studies will be needed to explore that relationship. We are now discussing this point in the discussion section.

Some questions that arise related to this:

Is the Lgr5+/CBC gene signature maintained in Ptk7-driven cystic organoids? Is the fetal ISC signature enriched? This would be expected based on the Yap activation. And, most importantly, what indication is there that Ptk7 plays any role in the intestine?

We thank the reviewer for these questions. We have found fetal ISC signature genes such as Anxa1 to be upregulated in SCM or Wnt5a treated organoids in our bulk RNAseq (Figure 6e and Supplementary figure 6a-d). We have now further performed snRNAseq from SCM-exposed organoids and find that, while cells with an Lgr5 signature do not show a significant change after SCM treatment, a population of cells expressing the fetal gene signature is increased after SCM exposure (Figure 1n-p and Supplementary Figure 6e-g).

With respect to evidence for a role for Ptk7 in the intestine, we are now including an analysis of Ptk7 expression by immunofluorescence, in situ hybridization, and qPCR, showing that Ptk7 is expressed in the intestinal epithelium and stroma (high in fibroblasts) (Figure 2g, h and Supplementary Figure 2e). In the aging intestine, we show using Western Blot that there is an increase in the secreted N-terminal domain of Ptk7 in the intestine (Figure 2i, j).

Ptk7 was identified from irradiated MEFs. While the study elegantly elucidates the effects of N-terminal Ptk7 on organoid morphology, there is no reason to believe that irradiate MEFs have anything to do with intestinal biology or a SASP program in that tissue. Is Ptk7 expressed in the epithelium or the subepithelial telocytes that provide Wnt ligands to the stem cells? Does Ptk7 N-terminal cleavage increase as a function of age in the intestine? Or in senescent cells in vivo? I think these critical points need to be addressed for a journal of this caliber.

We agree and have now included substantial new analysis to address these questions. We find that:

- (i) Ptk7 is expressed in intestinal epithelial and stromal cells (by immunofluorescence, in situ hybridization, and qPCR (Figure 2g, h and Supplementary Figure 2e).
- (ii) In the aging intestine, we find an increase in the secreted N-terminal domain of Ptk7 (by Western Blot - Figure 2i, j)
- (iii) We have now performed co-culture experiments of organoids with primary intestinal fibroblasts, showing that when senescence is induced in these cells, they induce a similar sPtk7-dependent cystic organoid phenotype (Figure 3i-k).

Minor comments:

-what is the expression of Ptk7 in MEFs vs. organoids? Can the authors show the ratio of total to cleaved N-terminal Ptk7, perhaps by Western?

Yes, we have now included the expression of Ptk7 in MEFs and organoids (Supplementary Figure 2c), as well as new results showing the ratio of full-length and cleaved Ptk7 by Western Blot (Figure 3i).

-what is the effect of Ptk7 treatment on long-term stem cell activity- e.g., organoids cultured in the presence of N-terminal Ptk7 digested down to single cells and replated serially over several passages would give an indication of the degree of stem cell self-renewal happening in the cultures.

We thank the reviewer for this question. We have now tested whether sPtk7 treatment causes a long-term change in colony forming activity of intestinal stem cells, using passaging of organoids as an assay (Supplementary Figure 1h, i). Consistent with our observation that withdrawal of SCM reverses the cystic organoid phenotype, we find no long-term impact of SCM on intestinal stem cell function. These results support our interpretation that sPtk7 causes a transient suppression of differentiation by activating Yap in intestinal epithelial cells.

-A number of studies have described the effects of Wnt5a/non-canonical Wnt signaling on the development of the intestinal epithelium. The findings in the current study should be discussed in relation to these prior studies.

We agree and have now included a discussion on non-canonical Wnt signaling in development and repair of the intestinal epithelium.

Collado, M., M. A. Blasco and M. Serrano (2007). "Cellular senescence in cancer and aging." Cell **130**(2): 223-233.

Demaria, M., N. Ohtani, S. A. Youssef, F. Rodier, W. Toussaint, J. R. Mitchell, R. M. Laberge, J. Vijg, H. Van Steeg, M. E. Dolle, J. H. Hoeijmakers, A. de Bruin, E. Hara and J. Campisi (2014). "An essential role for senescent cells in optimal wound healing through secretion of PDGF-AA." Dev Cell **31**(6): 722-733.

Gebert, N., C. W. Cheng, J. M. Kirkpatrick, D. Di Fraia, J. Yun, P. Schadel, S. Pace, G. B. Garside, O. Werz, K. L. Rudolph, H. Jasper, O. H. Yilmaz and A. Ori (2020). "Region-Specific Proteome Changes of the Intestinal Epithelium during Aging and Dietary Restriction." Cell Rep **31**(4): 107565.

Igarashi, M., M. Miura, E. Williams, F. Jaksch, T. Kadowaki, T. Yamauchi and L. Guarente (2019). "NAD(+) supplementation rejuvenates aged gut adult stem cells." Aging Cell **18**(3): e12935.

Imajo, M., M. Ebisuya and E. Nishida (2015). "Dual role of YAP and TAZ in renewal of the intestinal epithelium." Nat Cell Biol **17**(1): 7-19.

Jun, J. I. and L. F. Lau (2010). "The matricellular protein CCN1 induces fibroblast senescence and restricts fibrosis in cutaneous wound healing." Nat Cell Biol **12**(7): 676-685.

Mihaylova, M. M., C. W. Cheng, A. Q. Cao, S. Tripathi, M. D. Mana, K. E. Bauer-Rowe, M. Abu-Remaileh, L. Clavain, A. Erdemir, C. A. Lewis, E. Freinkman, A. S. Dickey, A. R. La Spada, Y. Huang, G. W. Bell, V. Deshpande, P. Carmeliet, P. Katajisto, D. M. Sabatini and O. H. Yilmaz (2018). "Fasting Activates Fatty Acid Oxidation to Enhance Intestinal Stem Cell Function during Homeostasis and Aging." Cell Stem Cell **22**(5): 769-778 e764.

Moorefield, E. C., S. F. Andres, R. E. Blue, L. Van Landeghem, A. T. Mah, M. A. Santoro and S. Ding (2017). "Aging effects on intestinal homeostasis associated with expansion and dysfunction of intestinal epithelial stem cells." Aging (Albany NY) **9**(8): 1898-1915.

Munoz-Espin, D. and M. Serrano (2014). "Cellular senescence: from physiology to pathology." Nat Rev Mol Cell Biol **15**(7): 482-496.

Nalapareddy, K., K. J. Nattamai, R. S. Kumar, R. Karns, K. A. Wikenheiser-Brokamp, L. L. Sampson, M. M. Mahe, N. Sundaram, M. B. Yacyshyn, B. Yacyshyn, M. A. Helmrath, Y. Zheng and H. Geiger (2017). "Canonical Wnt Signaling Ameliorates Aging of Intestinal Stem Cells." Cell Rep **18**(11): 2608-2621.

Pentinmikko, N., S. Iqbal, M. Mana, S. Andersson, A. B. Cognetta, 3rd, R. M. Suci, J. Roper, K. Luopajarvi, E. Markelin, S. Gopalakrishnan, O. P. Smolander, S. Naranjo, T. Saarinen, A. Juuti, K. Pietilainen, P. Auvinen, A. Ristimaki, N. Gupta, T. Tammela, T. Jacks, D. M. Sabatini, B. F. Cravatt, O. H. Yilmaz and P. Katajisto (2019). "Notum produced by Paneth cells attenuates regeneration of aged intestinal epithelium." Nature **571**(7765): 398-402.

Roulis, M., A. Kaklamanos, M. Scherthanner, P. Bielecki, J. Zhao, E. Kaffe, L. S. Frommelt, R. Qu, M. S. Knapp, A. Henriques, N. Chalkidi, V. Koliarakis, J. Jiao, J. R. Brewer, M. Bacher, H. N. Blackburn, X. Zhao, R. M. Breyer, V. Aidinis, D. Jain, B. Su, H. R. Herschman, Y. Kluger, G. Kollias and R. A. Flavell (2020). "Paracrine orchestration of intestinal tumorigenesis by a mesenchymal niche." Nature **580**(7804): 524-529.

Sirvinskas, D., O. Omrani, J. Lu, M. Rasa, A. Krepelova, L. Adam, S. Kaeppl, F. Sommer and F. Neri (2022). "Single-cell atlas of the aging mouse colon." iScience **25**(5): 104202.

Sovran, B., F. Hugenholtz, M. Elderman, A. A. Van Beek, K. Graversen, M. Huijskes, M. V. Boekschoten, H. F. J. Savelkoul, P. De Vos, J. Dekker and J. M. Wells (2019). "Age-associated Impairment of the Mucus Barrier Function is Associated with Profound Changes in Microbiota and Immunity." Sci Rep **9**(1): 1437.

Yui, S., L. Azzolin, M. Maimets, M. T. Pedersen, R. P. Fordham, S. L. Hansen, H. L. Larsen, J. Guiu, M. R. P. Alves, C. F. Rundsten, J. V. Johansen, Y. Li, C. D. Madsen, T. Nakamura, M. Watanabe, O. H. Nielsen, P. J. Schweiger, S. Piccolo and K. B. Jensen (2018). "YAP/TAZ-Dependent Reprogramming of Colonic Epithelium Links ECM Remodeling to Tissue Regeneration." Cell Stem Cell **22**(1): 35-49 e37.

Zhou, D., Y. Zhang, H. Wu, E. Barry, Y. Yin, E. Lawrence, D. Dawson, J. E. Willis, S. D. Markowitz, F. D. Camargo and J. Avruch (2011). "Mst1 and Mst2 protein kinases restrain intestinal stem cell proliferation and colonic tumorigenesis by inhibition of Yes-associated protein (Yap) overabundance." Proc Natl Acad Sci U S A **108**(49): E1312-1320.

REVIEWERS' COMMENTS

Reviewer #2 (Remarks to the Author):

The authors have improved the manuscript and adequately addressed my concerns.

Reviewer #3 (Remarks to the Author):

The authors have done a great job addressing my original comments. The only minor point that might be nice to address regards the analysis of the Yap-dependent fetal ISC signature and its relationship to the adult Wnt-Lgr5-dependent adult stem cell signature: 1) It would be nice to color code UMAPS based on enrichment for these signatures, rather than just expression of a couple of genes as a proxy for these signatures and 2) Based on the data presented and the literature surrounding the activation of the YAP signature during development and in adults during post-injury regeneration, the expectation is that cells in these two states should be mutually exclusive- i.e., one cell should not express both signatures concomitantly. I believe this to be the case in the scRNAseq data presented, but a direct visualization of this would be nice.

REVIEWERS' COMMENTS

Reviewer #2 (Remarks to the Author):

The authors have improved the manuscript and adequately addressed my concerns.

Reviewer #3 (Remarks to the Author):

The authors have done a great job addressing my original comments. The only minor point that might be nice to address regards the analysis of the Yap-dependent fetal ISC signature and its relationship to the adult Wnt-Lgr5-dependent adult stem cell signature: 1) It would be nice to color code UMAPS based on enrichment for these signatures, rather than just expression of a couple of genes as a proxy for these signatures and 2) Based on the data presented and the literature surrounding the activation of the YAP signature during development and in adults during post-injury regeneration, the expectation is that cells in these two states should be mutually exclusive- i.e., one cell should not express both signatures concomitantly. I believe this to be the case in the scRNAseq data presented, but a direct visualization of this would be nice.

We would like to thank the reviewers for their positive reception of our work.

- 1) We now include color code UMAPs based on ISC/CBC and YAP signature (Supplementary Figure 6e-h).
- 2) We also add UMAPs showing difference in ISC/CBC signature and YAP target genes and a graph showing the correlation between ISC/CBC signature and YAP target gene score (Supplementary Figure 6i and j).